# Breaking Training Bottlenecks: Effective Reinforcement Learning for Modern Coding Models

## Abstract

Modern code generation models exhibit longer outputs, accelerated capability growth, and fundamentally changed training dynamics, rendering traditional training methodologies, algorithms, and datasets ineffective for enhancing their performance. To address these training bottlenecks, we propose **MicroCoder-GRPO**, an enhanced Group Relative Policy Optimization approach with three key innovations: conditional truncation masking to enhance long output potential while maintaining training stability, diversity-determined temperature selection to maintain and encourage output diversity, and removal of KL loss with high clipping ratios to facilitate exploration. MicroCoder-GRPO achieves up to 17.6% relative improvement over strong baselines on LiveCodeBench v6, with more pronounced gains under extended context evaluation. Additionally, we release **MicroCoder-Dataset**, a more challenging training corpus that achieves 3× larger performance gains than mainstream datasets on LiveCodeBench v6 within 300 training steps, and **MicroCoder-Evaluator**, a robust framework with approximately 25% improved evaluation accuracy and around 40% faster execution. Through comprehensive analysis across more than thirty controlled experiments, we reveal 34 key **training insights** across seven main aspects, demonstrating that properly trained models can achieve competitive performance with larger counterparts.

## 1 Introduction

### 1.1 Background & Related Work

Scaling inference time through extended reasoning enables models to solve increasingly challenging problems, while reinforcement learning has proven effective at activating reasoning capabilities and knowledge acquired during pretraining, even without instruction fine-tuning. Group Relative Policy Optimization (GRPO) (Shao et al., 2024) has gained attention by eliminating the value model requirement, instead sampling multiple responses per problem and computing relative advantages. Recent GRPO algorithmic improvements have been primarily validated on mathematical reasoning tasks, addressing specific optimization limitations. Dr. GRPO (Liu et al., 2025b) identified that GRPO encourages shorter correct responses and longer incorrect ones, leading to modifications that remove token-level averaging in loss computation and reward standard deviation normalization in advantage calculation, thus limiting output length growth. DAPO (Yu et al., 2025) enhanced exploration by eliminating KL divergence loss and employing high clipping ratios. Polaris (An et al., 2025) analyzed the effects of different training components such as temperature and dataset difficulty on mathematical task training.

Reinforcement learning for code generation has evolved through several key approaches, beginning with CodeRL (Le et al., 2022) applying the REINFORCE algorithm for offline training with program-level rewards. PPOCoder (Shojaee et al., 2023) improved upon this by adopting PPO optimization with online training, while RLTF (Liu et al., 2023) eliminated the value model and introduced both coarse-grained rewards (overall code pass/timeout/syntax errors) and fine-grained rewards (specific error line feedback) within an online training framework. StepCoder (Dou et al., 2024) employed PPO with curriculum learning, initially providing partial solutions and progressively reducing assistance while updating only tokens corresponding to executed code. SRPO

(Zhang et al., 2025), built on GRPO, analyzed both mathematical and coding tasks, observing that mathematical problems tend to increase output length while coding problems tend to decrease it, though our findings indicate that newer model generations like Qwen-3 exhibit length growth tendencies even for coding tasks. The field has been supported by open-source projects including DeepCoder (Luo et al., 2025), Open-R1 (Hugging Face, 2025), and Code-R1 (Liu & Zhang, 2025), alongside integrated training datasets such as Taco (Li et al., 2023), KodCode (Xu et al., 2025), and rStar-Coder (Liu et al., 2025a). However, research on GRPO applications to coding tasks remains relatively limited compared to mathematical reasoning domains.

## 1.2 MOTIVATION

Reinforcement learning insights for code generation differ from those for mathematical tasks, as coding problems require passing all test cases with additional conditions such as runtime limitations, making them more challenging and complex. Furthermore, previously accumulated training insights and datasets for traditional models often prove ineffective for modern models that have extended outputs and high reasoning abilities. Therefore, effective training of modern coding models requires updated and more challenging datasets of higher quality, new algorithms that unlock long output potential, and fresh training insights.

For example, training with GRPO on the mainstream DeepCoder dataset shows substantial improvements in Qwen 2.5 models (Qwen et al., 2025) but minimal improvements in Qwen 3 models (Yang et al., 2025), as shown in Figure 1. Critic reward analysis reveals that mainstream datasets pose greater difficulty for Qwen 2.5 while appearing relatively simple for Qwen 3 capabilities. Output behaviors reveal fundamental generational differences. Qwen 3 models exhibit pronounced upward trends in response length during training, whereas Qwen 2.5 models show stable or decreasing lengths. Additionally, across the model series progression, standard model outputs show increasing length and variance from Qwen 2.5-Instruct to Qwen 3-Instruct to Qwen 3-Thinking.

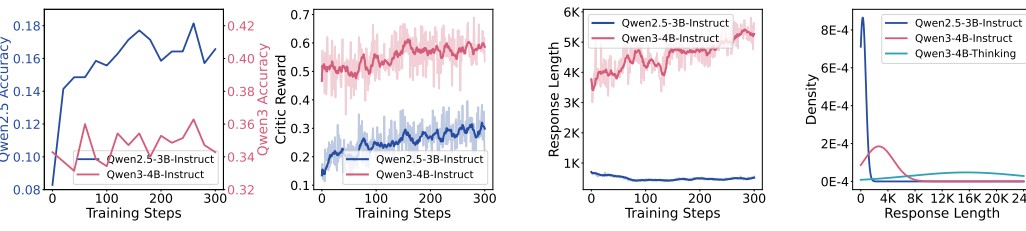

Figure 1: **Cross-Model Training Dynamics**. Performance and response length across Qwen 2.5 and Qwen 3 models, illustrating generation-specific training behaviors and output characteristics.

## 1.3 CONTRIBUTIONS

This paper makes four key contributions to improve reinforcement learning for coding models.

- **Algorithmic Innovation**: We propose MicroCoder-GRPO, extending GRPO with conditional truncation masking, diversity-determined temperature selection, and removal of KL loss with high clipping. This approach achieves up to 17.6% relative improvement over strong baselines on LiveCodeBench v6, demonstrating robust performance gains across multiple model scales.
- **Systematic Analysis**: Through over thirty controlled experiments, we provide comprehensive analysis of important training components including dataset quality, code evaluators, temperature dynamics, context length and extension, truncation masking strategies, batch size and on-policy, and KL and clip ratio, offering detailed insights into their effects on reinforcement learning for code generation.
- **Dataset Creation**: We introduce MicroCoder-Dataset, a higher-quality and more challenging training corpus that achieves 3× larger performance gains than the DeepCoder dataset on Live-CodeBench v6 within 300 training steps, demonstrating effectiveness for developing coding capabilities in modern language models.
- **Infrastructure Development**: We design MicroCoder-Evaluator, a robust evaluation framework that improves evaluation accuracy by approximately 25% compared to the LiveCodeBench code

evaluator while achieving around 40% faster execution per training step through optimized parallel processing compared, enabling more reliable training feedback and improved computational efficiency.

## 2 ALGORITHMS

### 2.1 PRELIMINARIES: GRPO

Group Relative Policy Optimization (GRPO) reduces reinforcement learning training costs by eliminating the separate value model typically required in policy gradient methods. For each query $q$, GRPO samples a group of $G$ outputs $\{o_1, o_2, \ldots, o_G\}$ from the reference policy $\pi_{\theta_{old}}$ and optimizes the current policy $\pi_\theta$ by maximizing $\mathcal{J}_{GRPO}(\theta)$. The advantage function $A_i$ employs group-based normalization to estimate relative output quality. GRPO is shown as the black components in Equations 1 and 2.

### 2.2 MICROCODER-GRPO

We propose MicroCoder-GRPO, which introduces three main modifications to standard GRPO. First, conditional truncation masking selectively zeros advantage scores for responses that simultaneously reach maximum length $L_{max}$, produce incorrect answers, avoid repetition sequences (final 128 tokens differ from preceding 128 tokens), and are randomly selected with probability $\rho$. Second, diversity-determined temperature selection determines training temperature $T(D)$ based on the model's initial output diversity values and subsequent trends, ensuring the chosen temperature prevents rapid and sustained diversity decline that leads to training failure. Third, following DAPO, we remove KL loss ($\beta_0 = 0$) and employ high clipping ($\varepsilon_{high}$) to maintain output diversity and response length growth. The modifications of MicroCoder-GRPO compared to GRPO are shown as the red components in the equations:

$$\mathcal{J}_{GRPO}(\theta) = \mathbb{E}_{q \sim P(Q)} \left[ \sum_{i=1}^{G} \left( \min \left( \frac{\pi_\theta^{T(D)}(o_i|q)}{\pi_{\theta_{old}}^{T(D)}(o_i|q)} A_i, \text{clip}\left( \frac{\pi_\theta^{T(D)}(o_i|q)}{\pi_{\theta_{old}}^{T(D)}(o_i|q)}, 1 - \varepsilon, 1 + \varepsilon_{high} \right) A_i \right) - \beta_0 \mathbf{D}_{KL}\left( \pi_\theta \| \pi_{\theta_{old}} \right) \right) \right] \quad (1)$$

$$A_i = \frac{r_i - \text{mean}(\{r_1, r_2, \ldots, r_G\})}{\text{std}(\{r_1, r_2, \ldots, r_G\})} \cdot (1 - \mathbb{I}\left[ |o_i| = L_{max} \wedge \text{incorrect}(o_i) \wedge \neg\text{repeat}(o_i, m) \wedge U(0,1) < \rho \right]) \quad (2)$$

where $\theta$ and $\theta_{old}$ represent current and reference policy parameters, $\varepsilon$ controls the clipping trust region, $\beta$ weights KL divergence regularization, $r_i$ denotes the reward for output $o_i$, incorrect$(o_i)$ indicates whether output $o_i$ is incorrect, $\neg$repeat$(o_i, m)$ checks for non-repetition sequences, $\rho$ controls masking probability, $U(0,1)$ denotes uniform distribution over $[0,1]$, and $\mathbb{I}[\cdot]$ is the indicator function.

### 2.3 CONDITIONAL TRUNCATION MASK

Truncation masking enhances long output generation potential by setting advantage scores to zero for responses that reach maximum length limits, preventing truncated outputs from contributing to policy optimization. Conditional truncation masking applies selective criteria, only masking responses that simultaneously reach maximum length, achieve correct answers, avoid repetition sequences (final 128 tokens differ from preceding 128 tokens), and masks only a specified proportion rather than all qualifying responses. As shown in Figure 2, higher masking rates accelerate output length growth and push convergence values closer to maximum response limits, with 30% masking achieving growth rates comparable to complete masking. Increased masking also accelerates response diversity decline and reduces diversity convergence values.

Masking creates distinct performance dynamics where training rapidly rises to higher values, then declines, and converges to specific performance levels. Masking proportion creates trade-offs between training speed and peak performance. Increased masking enables faster achievement of initial performance peaks, while reduced masking extends the initial improvement phase and achieves higher peak performance values. Conditional truncation masking demonstrates superior training stability compared to both no masking and complete masking approaches, achieving higher final performance while avoiding the rapid training decrease observed with complete masking strategies.

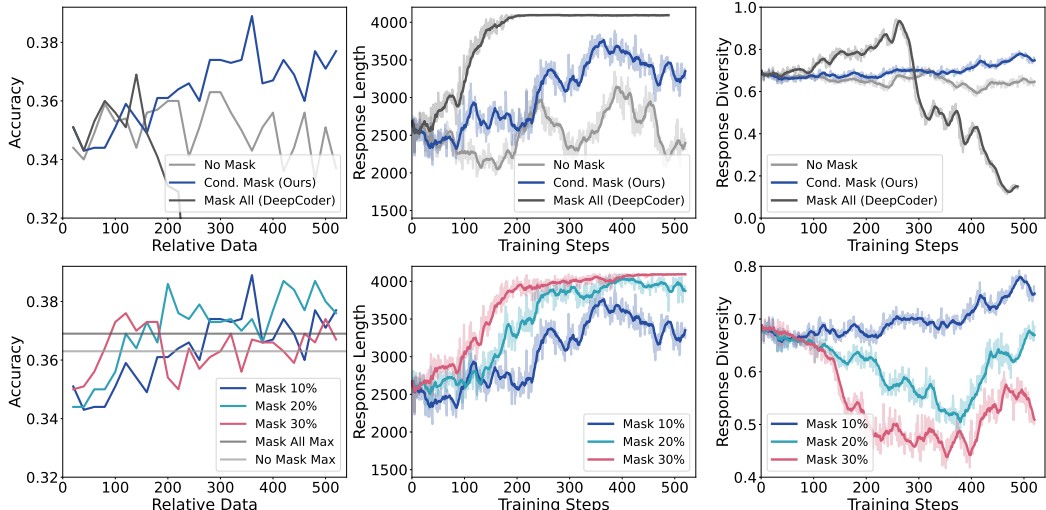

Figure 2: **Truncation Masking Effects on Training.** Performance trends under different masking strategies, comparing no mask, complete masking, and conditional masking at various rates.

## 2.4 DIVERSITY-DETERMINED TEMPERATURE

Temperature scheduling influences GRPO training stability and convergence (Figure 3). Analysis reveals that models develop increasing temperature robustness throughout training, with the upper bound of stable temperatures progressively expanding. Higher temperatures naturally increase output diversity, but this diversity systematically decreases at fixed temperatures as training progresses. Despite varying temperatures, output diversity converges to similar final values across different temperature settings. Important temperature thresholds emerge during training. When initial output diversity falls below expected convergence values, models experience continued diversity reduction accompanied by training failure. Traditionally standard temperatures (t=0.6) can cause training failure, while modern models like Qwen-3 demonstrate stable training even at elevated temperatures (t=1.8) with minimal influence on final convergence values. Therefore, training temperature can be determined based on response diversity, selecting values that avoid both excessively low temperatures causing continuous decline in output diversity and excessively high temperatures leading to drastic fluctuations, with optimal temperatures enabling stable diversity convergence.

Dynamic temperature scheduling yields superior performance compared to static temperature approaches. The optimal strategy employs initial low-temperature training followed by high-temperature phases. This approach reduces initial output diversity during the high-temperature stage, ultimately achieving better performance than direct high-temperature training from initialization. However, we observe that continuous uniform temperature changes influence training stability, and even brief continuous temperature increases or decreases within a small number of steps can cause irreversible change in output diversity. Therefore, this paper employs staged temperature transitions or determines the initial constant training temperature based on output diversity.

## 2.5 NO KL LOSS AND HIGH CLIP RATIO

Removing KL loss with high clipping enhances output diversity and response length, driving sustained performance improvements (Figure 4). Standard KL loss without high clipping reduces output diversity and limits response length to marginal increases, causing initial performance gains followed by decline. Continued diversity reduction creates unsustainable training dynamics where performance first rises then falls, preventing effective long-term training.

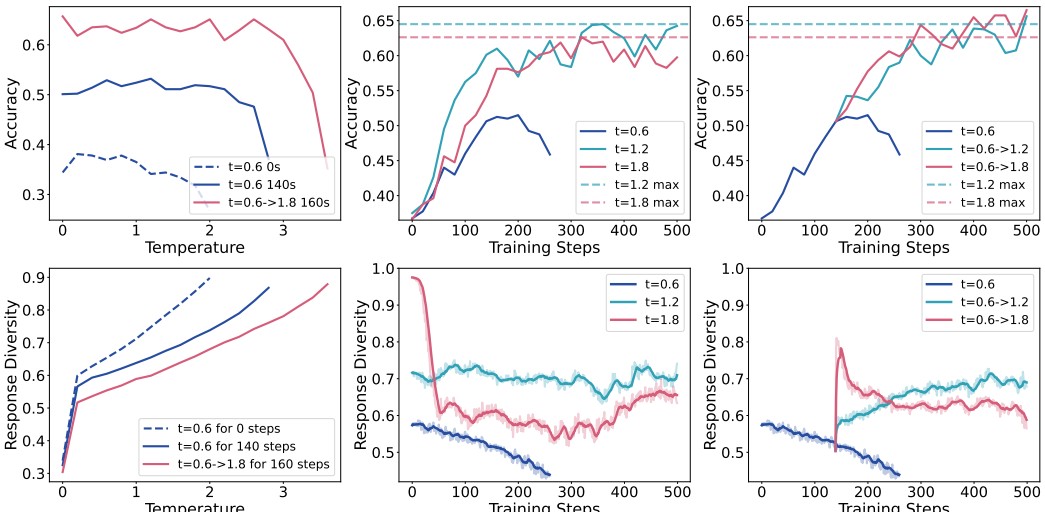

Figure 3: **Temperature Dynamics and Scheduling Analysis.** Temperature robustness trends during training showing increasing stability at higher temperatures, convergence of output diversity across different temperature settings, training failure when low temperatures cause initial diversity to fall below convergence values, and superior performance of dynamic temperature scheduling (low-to-high transition) compared to static temperature approaches.

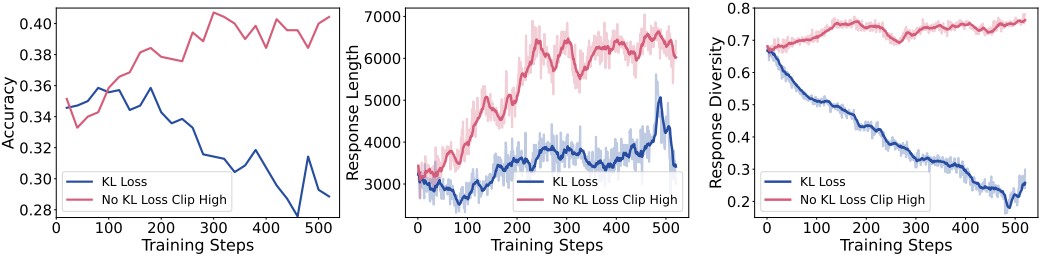

Figure 4: **Influence of KL Loss and Clip Ratio on Training Dynamics**. Performance comparison between standard KL loss and removed KL loss with high clipping, illustrating the relationship between diversity maintenance and sustained performance improvement.

## 3 DATA

Our data processing framework employs a four-stage pipeline to create high-quality coding datasets. The Collect stage aggregates data from diverse sources to maximize coverage. The Process stage standardizes data through language translation, noise removal, format normalization, and completeness validation. The Filter stage applies multi-criteria selection based on textual quality, content relevance, and difficulty distribution. Finally, the Verify stage conducts validation to ensure problem readability, completeness, and test case accuracy. This end-to-end pipeline transforms raw datasets into a high-quality, standardized corpus suitable for reinforcement learning training.

Comparative analysis reveals that MicroCoder dataset drives superior coding ability improvements over DeepCoder dataset (Figure 5). Training on MicroCoder dataset yields rapid, obvious accuracy gains, while DeepCoder dataset training shows minimal performance variation. This performance differential correlates with inherent dataset difficulty. MicroCoder dataset consistently generates lower critic rewards, indicating higher problem complexity. Despite both datasets exhibiting similar critic reward growth trends during training, only MicroCoder dataset produces test set improvements, demonstrating that training effectiveness on challenging problems translates more directly to generalization performance.

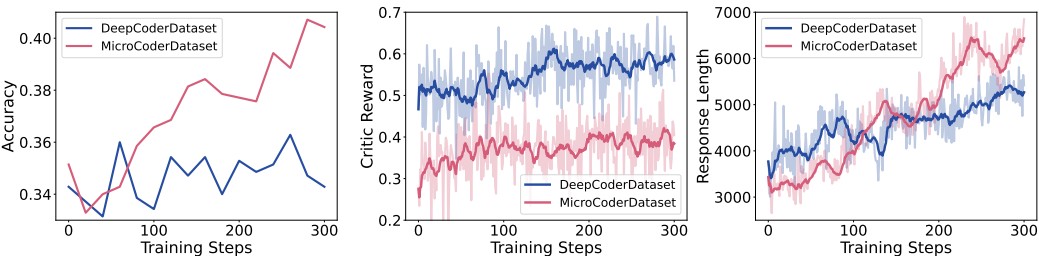

Figure 5: **Dataset Quality Comparison.** Training dynamics comparing MicroCoder and Deep-Coder datasets across accuracy, critic reward, and response length metrics, demonstrating learning effectiveness on challenging problems.

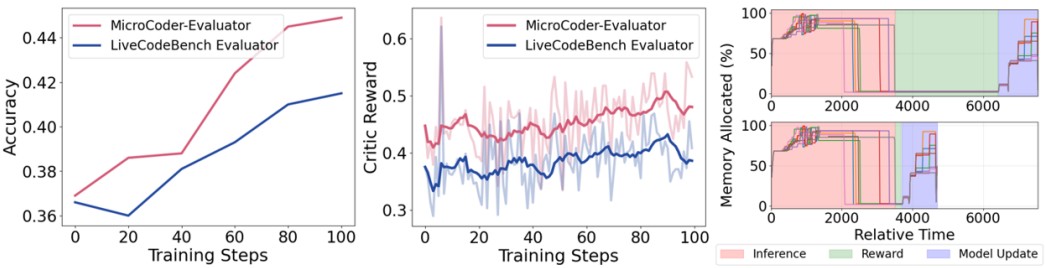

Figure 6: **Training Performance with Different Code Evaluators.** Accuracy and reward during training using MicroCoder-Evaluator versus LiveCodeBench Evaluator, demonstrating the benefits of robust output validation for coding task training. The right subfigure shows the efficiency improvements of MicroCoder-Evaluator through parallel processing optimization compared to the original DeepCoder single-threaded version.

Harder problems exhibit accelerated response length growth with greater final magnitudes. Although MicroCoder dataset initially produces similar or shorter response lengths compared to DeepCoder dataset, it demonstrates faster growth rates and ultimately achieves longer outputs. This indicates that challenging coding tasks require longer solution paths.

## 4 INFRASTRUCTURE

MicroCoder-Evaluator implements comprehensive output validation through multi-strategy comparison with 6-7 fallback methods, format flexibility handling lists, tuples, strings, sets with automatic type conversions, approximate numeric comparison using np.allclose() for floating point tolerance, preprocessing including multi-line splitting and whitespace normalization, and high fault tolerance that continues attempting different comparison approaches when individual methods fail. In contrast, LiveCodeBench Evaluator employs exact matching through direct equality comparison, precise numerics via Decimal library for high-precision floating point operations, and minimal preprocessing limited to basic whitespace stripping.

The comprehensive evaluation approach yields superior training outcomes (Figure 6). Higher accuracy values generally represent more reliable gold standard evaluation, as comprehensive comparison methods better capture valid solution variations when matching outputs against ground truth answers. MicroCoder-Evaluator achieves higher critic reward scores, indicating more accurate assessment of solution quality. This translates to enhanced model training effectiveness with fewer misjudgments, reduced noise injection, and accelerated and higher test accuracy improvement. The performance differential between evaluators is particularly pronounced during early training stages, where robust evaluation becomes important for establishing proper learning feedback and preventing suboptimal convergence.

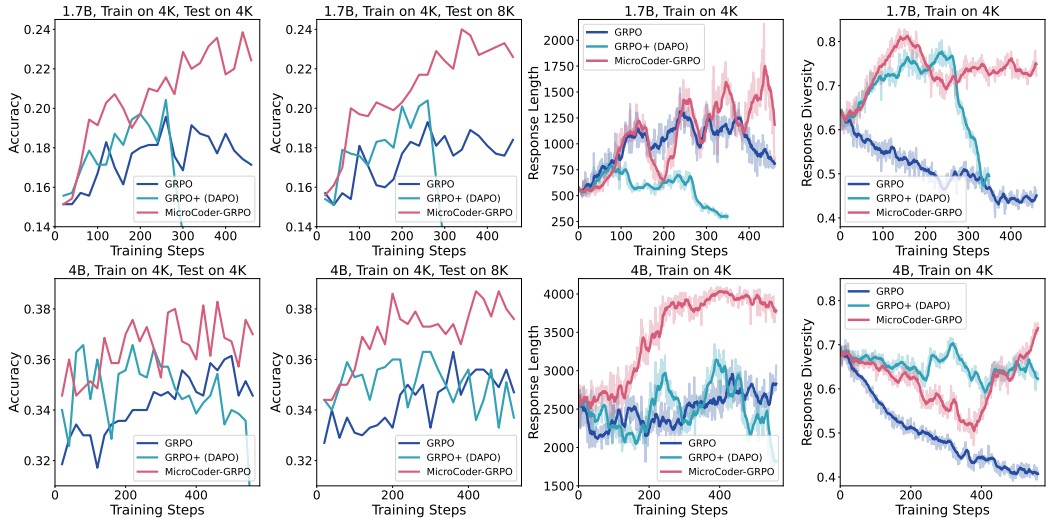

Figure 7: **Illustrative performance comparison** between MicroCoder-GRPO and baseline approaches across different model scales and output lengths, illustrating training accuracy, response length, and output diversity dynamics, demonstrating the superior stability and sustained improvements of the proposed method under both preset and extended context lengths.

## 5 EXPERIMENTAL DESIGN

For algorithmic comparisons, we evaluate MicroCoder-GRPO against GRPO (Shao et al., 2024) and GRPO+ (DAPO) (Yu et al., 2025) that removes KL loss and employ high clipping. For temperature dynamics analysis, we employ the Open-R1 (Hugging Face, 2025) training set with evaluation on 200 randomly selected problems from the test set using maximum response length of 4K tokens. All other analyses and model evaluations utilize DeepCoder (Luo et al., 2025) and MicroCoder datasets with testing conducted on the unseen AtCoder and LeetCode in LiveCodeBench v6. Experiments are performed using Qwen3-1.7B Instruct and Qwen3-4B-Instruct-2507 models (Yang et al., 2025) to demonstrate method robustness across different model scales. Unless otherwise specified, default experimental configurations employ MicroCoder dataset, LiveCodeBench v6 evaluation, Qwen3-4B-Instruct-2507 model, maximum response length of 8K tokens, temperature of 1.2, train batch size of 64, learning rate of 1e-6, GRPO+ algorithm, 8 samples per query during training with 0-1 binary accuracy as reward, and evaluation based on average accuracy across four inference attempts.

## 6 RESULTS

Experimental evaluation compares training dynamics under 4K response length with testing on both 4K and 8K contexts to assess algorithmic effectiveness and reasoning budget scalability (Figure 7). MicroCoder-GRPO demonstrates superior performance across both testing configurations, achieving faster convergence rates and higher final accuracy values compared to baselines. Additionally, our 4K context training achieves performance comparable to baseline methods trained with 6K contexts, while saving approximately 40-50% computational cost due to the O(n²) of self-attention. While GRPO+ (DAPO) with removed KL loss and high clipping reaches higher peak performance than standard GRPO and achieves peak values more rapidly, it exhibits training instability with pronounced performance decrease during extended training phases. In contrast, MicroCoder-GRPO's conditional truncation masking not only accelerates improvement and enhances convergence values but maintains stable long-term training dynamics without the failure observed in GRPO+. Analysis reveals that 4B models exhibit greater response length growth capacity compared to 1.7B models, with MicroCoder-GRPO producing length increases across both model scales while preserving output diversity and sustained accuracy improvements throughout training.

MicroCoder-GRPO consistently outperforms both GRPO and GRPO+ (DAPO) across all benchmark datasets, difficulty levels, and model sizes, demonstrating robust improvements in coding

| | AtCoder | | | | LeetCode | | | | LiveCodeBench | | | |
|---|---|---|---|---|---|---|---|---|---|---|---|---|
| | Easy | Medium | Hard | All | Easy | Medium | Hard | All | Easy | Medium | Hard | All |
| *Not trained* | | | | | | | | | | | | |
| Qwen3-1.7B Instruct (4K) | 69.2 | 10.6 | 1.3 | 19.2 | 33.8 | 3.8 | 0 | 10.7 | 55.2 | 7.2 | 0.9 | 16.1 |
| Qwen3-1.7B Instruct (8K) | 69.2 | 10.6 | 1.3 | 19.2 | 33.8 | 3.8 | 0 | 10.7 | 55.2 | 7.2 | 0.9 | 16.1 |
| Qwen3-4B-Instruct (4K) | 95.2 | 44.2 | 9.2 | 37.3 | 73.5 | 19.2 | 2.5 | 28.6 | 86.6 | 31.7 | 7.5 | 34.1 |
| Qwen3-4B-Instruct (8K) | 95.2 | 45.2 | 10.0 | 37.9 | 76.5 | 18.3 | 2.5 | 29.0 | 87.8 | 31.7 | 8.1 | 34.7 |
| *1.7B, Train on 4K, Test on 4K* | | | | | | | | | | | | |
| GRPO | 70.2 | 20.2 | 1.7 | 21.9 | 47.1 | **4.8** | **0** | 14.7 | 61.0 | 12.5 | 1.3 | 19.3 |
| GRPO+ (DAPO) | 76.9 | 15.4 | 2.9 | 23.0 | 55.9 | 1.9 | **0** | 15.9 | 68.6 | 8.7 | **2.2** | 20.4 |
| **MicroCoder-GRPO** | **83.7** | **24.0** | **2.9** | **26.6** | **57.4** | **4.8** | **0** | **17.5** | **73.3** | **14.4** | **2.2** | **23.3** |
| Δ | +6.8 | +3.8 | 0.0 | +3.6 | +1.5 | 0.0 | 0 | +1.6 | +4.7 | +1.9 | 0.0 | +2.9 |
| *1.7B, Train on 4K, Test on 8K* | | | | | | | | | | | | |
| GRPO | 70.2 | 20.2 | 1.7 | 21.9 | 47.1 | 4.8 | **0** | 14.7 | 61.0 | 12.5 | 1.3 | 19.3 |
| GRPO+ (DAPO) | 76.9 | 15.4 | 2.9 | 23.0 | **55.9** | 1.9 | **0** | 15.9 | 68.6 | 8.7 | 2.2 | 20.4 |
| **MicroCoder-GRPO** | 82.7 | 26.9 | 3.7 | 27.5 | 52.9 | 8.7 | **0** | 17.9 | 70.9 | 17.8 | 2.8 | 24.0 |
| Δ | +5.8 | +6.7 | +0.8 | +4.5 | -3.0 | +3.9 | 0 | +2.0 | +2.3 | +5.3 | +0.6 | +3.6 |
| *4B, Train on 4K, Test on 4K* | | | | | | | | | | | | |
| GRPO | 98.1 | 46.2 | **11.7** | **39.7** | 72.1 | 21.2 | 0 | 28.2 | 87.8 | 33.7 | **8.7** | 35.6 |
| GRPO+ (DAPO) | 97.1 | **49.0** | 10.0 | 39.3 | 75.0 | **22.1** | 1.3 | 29.8 | 88.4 | **35.6** | 7.8 | 35.9 |
| **MicroCoder-GRPO** | **100.0** | **49.0** | 9.6 | **39.7** | **88.2** | **22.1** | **3.7** | **34.1** | **95.3** | **35.6** | 8.1 | **37.7** |
| Δ | +1.9 | 0.0 | -2.1 | 0.0 | +13.2 | 0.0 | +2.4 | +4.3 | +6.9 | 0.0 | -0.6 | +1.8 |
| *4B, Train on 4K, Test on 8K* | | | | | | | | | | | | |
| GRPO | **100.0** | 45.2 | **12.9** | 40.6 | 72.1 | 22.1 | 0 | 28.6 | 89.0 | 33.7 | 9.7 | 36.3 |
| GRPO+ (DAPO) | **100.0** | 43.3 | 11.3 | 39.3 | 76.5 | 23.1 | 2.5 | 31.0 | 90.7 | 33.2 | 9.1 | 36.3 |
| **MicroCoder-GRPO** | **100.0** | **48.1** | **12.9** | **41.3** | **83.8** | **24.0** | **5.0** | **34.1** | **93.6** | **36.1** | **10.9** | **38.7** |
| Δ | 0.0 | +2.9 | 0.0 | +0.7 | +7.3 | +0.9 | +2.5 | +3.1 | +2.9 | +2.4 | +1.2 | +2.4 |

Table 1: **Quantitative evaluation results** across different datasets comparing MicroCoder-GRPO against baseline methods on various model sizes, difficulty levels, and output budgets to assess algorithmic improvements and extended reasoning capabilities.

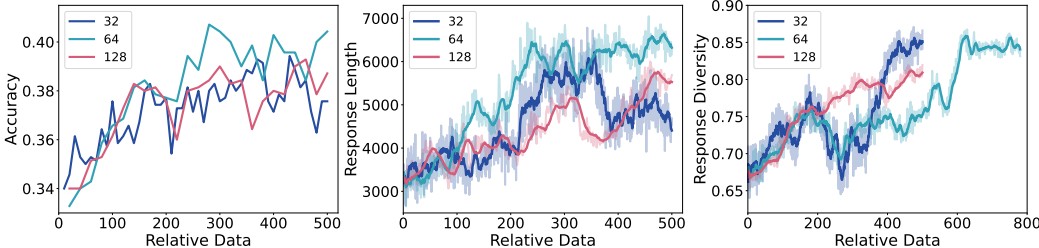

Figure 8: **On-Policy versus Off-Policy Training Effects**. Performance comparison across different train batch size settings, demonstrating that off-policy configurations increase training stability while intermediate settings achieve optimal performance.

task performance (Table 1). The performance gains become more pronounced under extended context evaluation, where models trained on 4K contexts are tested on 8K contexts, with 1.7B models achieving +3.6% improvement on LiveCodeBench, indicating superior scalability of the proposed approach. Harder problems exhibit greater performance gains under extended output budgets, with MicroCoder-GRPO showing strong advantages on Medium and Hard difficulty levels, suggesting enhanced capability for complex reasoning tasks requiring longer solution development.

# 7 ANALYSIS

## 7.1 BATCH SIZE AND ON-POLICY

Training batch configuration influences on-policy versus off-policy learning characteristics. The framework employs train_batch_size for simultaneous problem inference and ppo_mini_batch_size for individual parameter updates, executing train_batch_size/ppo_mini_batch_size update iterations per training step cycle. Smaller train_batch_size values (maintaining constant ppo_mini_batch_size) create more on-policy behavior resembling immediate problem-solving reflection, while larger values produce off-policy dynamics akin to batch reflection after completing all problems. As shown in Figure 8, on-policy configurations exhibit reduced training stability with accelerated response diversity convergence and response length trends that rise and then decline, whereas off-policy approaches demonstrate greater stability across both metrics. Optimal performance emerges from in-

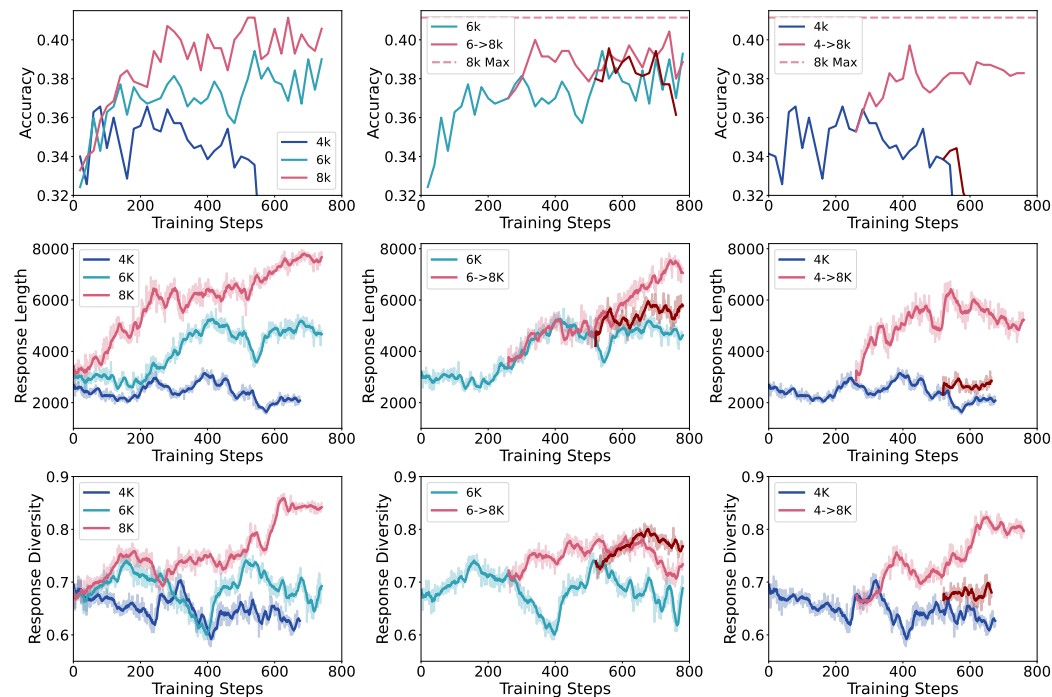

Figure 9: **Context Length Influence on Training Dynamics.** Performance trends across different maximum output length settings showing accuracy, response length growth, and output diversity, demonstrating irreversible effects of early-stage length limitations on model capabilities.

termediate configurations balancing on-policy and off-policy characteristics, outperforming heavily skewed settings in either direction.

## 7.2 CONTEXT LENGTH AND EXTENSION

Context length settings exhibit scaling relationships with model performance (Figure 9). Longer maximum output lengths correlate with higher final accuracy, faster output growth rates, greater final output lengths, and increased output diversity. However, small initial maximum output lengths reduce both output generation and diversity, creating persistent performance effects even after subsequent length extensions. The magnitude of these effects increases with more limiting initial settings and longer training time. Beyond specific training thresholds, models show minimal recovery when limitations are later relaxed, indicating that early-stage output reduction fundamentally changes learning paths and cannot be compensated by later context extension.

## 8 CONCLUSIONS

To address the training bottlenecks of modern code generation models, this paper proposes MicroCoder-GRPO, a novel algorithm with three key innovations: conditional truncation masking to enhance long output potential while maintaining training stability, diversity-determined temperature selection to maintain output variability, and removal of KL loss with high clipping ratios to encourage exploration. We introduce a more challenging and higher-quality dataset, and develop a more robust code evaluator with faster testing speed. Through comprehensive analysis of over thirty controlled experiments, we derive 34 important training insights (Sections 2, 3, 4, and 7) that provide systematic guidance for reinforcement learning in code generation.

Future work will focus on applying our methods and insights to diverse coding tasks, fully leveraging the performance improvements, high efficiency, and strong generalization capabilities demonstrated in this work. The systematic approach established here opens new possibilities for advancing reinforcement learning across various code generation domains.

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

## A  APPENDIX

### A.1  THE USE OF LLMS

LLMs are used to check the grammar and spelling of this paper.

