# OpenReview forum: "Breaking Training Bottlenecks: Effective Reinforcement Learning for Modern Coding Models"
_ICLR.cc/2026/Conference — Submitted to ICLR 2026_

### Official Review · Reviewer_HK1x · 2025-10-25

**Soundness:** 2
**Presentation:** 3
**Contribution:** 3
**Rating:** 6
**Confidence:** 2

**Summary:**

This paper proposes MicroCoder-GRPO, an improved reinforcement learning optimization framework for code generation tasks based on Group Relative Policy Optimization (GRPO). The method introduces three main innovations: Conditional Truncation Masking, Diversity-Determined Temperature Selection, and Removal of KL loss with high clipping. Additionally, the paper presents MicroCoder-Evaluator. Experimental results demonstrate that MicroCoder-GRPO achieves higher training stability, sustained diversity, and improved accuracy compared with baseline GRPO and LiveCodeBench-based training setups.

**Strengths:**

1. The paper clearly identifies new training bottlenecks specific to modern coding models (long outputs, altered dynamics, and reasoning-heavy behavior) that traditional RL methods fail to address.

2. The proposed MicroCoder-GRPO introduces three complementary and practically effective modifications (conditional truncation masking, diversity-determined temperature scheduling, and removal of KL loss with high clipping), each addressing a concrete limitation in prior GRPO-style training.

3. The MicroCoder dataset and MicroCoder-Evaluator represent well-designed, practically valuable resources for future RL-based code generation research.

**Weaknesses:**

1. The proposed modifications (conditional truncation masking, diversity-based temperature scheduling, KL removal with high clipping) are primarily empirically motivated. The paper lacks formal justification or convergence analysis to establish why these strategies work beyond observed trends.

2. Experiments are conducted exclusively on Qwen-based models and LiveCodeBench-derived datasets, raising concerns about generalizability to other architectures.

3. While the combination of improvements is effective, each modification represents a relatively straightforward heuristic extension of prior GRPO/DAPO frameworks.

4. The claim of faster convergence and efficiency is described qualitatively; detailed wall-clock time, GPU cost, or training throughput comparisons to GRPO/DAPO are unclear.

**Questions:**

1. Could the authors elaborate on how temperature is dynamically updated (formula or algorithmic rule) and whether the same policy generalizes across different model sizes?

2. Have the authors tested MicroCoder-GRPO on other architectures (e.g., CodeLlama, DeepSeek-Coder, or StarCoder)? If not, what aspects of the algorithm might depend on the Qwen-specific tokenizer or output behavior?

3. can the authors clarify how much of the final gain (17.6%) originates from the new algorithm versus the new dataset?

4. Are there particular coding domains or problem types where MicroCoder-GRPO underperforms or shows instability compared to GRPO?

---

> ### Author Response · Authors · 2025-11-20
> **Response 1/7**
>
> Dear Reviewer,
>
> We deeply appreciate your thorough and useful feedback. Your comments demonstrate significant expertise in this field and have greatly improved the rigor and comprehensiveness of our manuscript.
>
> We have meticulously addressed each of your concerns with detailed responses and additional analysis provided below.
>
> > ## **W2, Q2: Model Generalizability (Part 1):**
>
> - **Reasons for Choosing Qwen-3 Models:**
>   - **State-of-the-Art Open-Source** Models
>     - Qwen-3 represents the **latest** generation of open-source state-of-the-art code generation models at the time of our study
>     - This positions our work to address **training challenges** most relevant to current and future model development
>   - Exhibits **Distinct Characteristics** from Previous Model Generations
>     - Our motivation stems from observing **fundamental differences** between Qwen-3 and earlier models (including Qwen-2.5):
>     - Significantly **longer average output length**
>       - Qwen-3 base output: **~4K** tokens
>       - Previous generation (e.g., DeepSeek-distill-Qwen): **~1K** tokens
>       - This represents a **4×** increase in generation capacity
>     - More pronounced **length scaling during RL training**
>       - As shown in Figure 1, Qwen-3 exhibits **clear upward trends** in response length during training
>       - In contrast, Qwen-2.5 shows **stable or even decreasing** length trends
>       - This **fundamental difference** motivated our investigation into training methods for **length-scaling models**
>     - Supports **both** long-chain-of-thought and standard reasoning modes
>       - Qwen-3 family includes **both** Instruct and Thinking variants
>       - Enables studying **different reasoning methods** within the same architectural family
>     - Represents the **core motivation** of our work
>       - Our paper explicitly aims to study:
>         - **Extended output** capabilities
>         - Breaking through **capability bottlenecks**
>         - Fundamentally changed **training dynamics**
>       - Qwen-3 embodies these modern characteristics, making it the ideal testbed for our research questions
>
> - **Why Not DeepSeek-Distill-Qwen and Other Earlier Models:**
>   - We considered but decided against using **earlier models** as our primary experimental platform for several principled reasons
>   - **Similar characteristics** to **Qwen-2.5** generation
>     - DeepSeek-distill-Qwen and similar models exhibit training dynamics more **similar** to **Qwen-2.5** than to Qwen-3
>     - They do not demonstrate the **length-scaling phenomena** that motivated our work
>   - Much **shorter output lengths**
>     - Typical output: **~1K** tokens (vs. 4K for Qwen-3)
>     - This **4× difference** means these models do not face the same truncation and length management challenges that our methods address
>   - **Decreasing** rather than increasing length during **RL training**
>     - Earlier models show **stable or decreasing** response length during RL on coding tasks
>     - This contradicts the **length-growth trend** in modern models like Qwen-3
>   - **Capability gap** with modern models
>     - There exists a notable **performance gap** between earlier distilled models and Qwen-3
>     - Studying older models would not address the **training challenges** posed by more capable recent architectures
>   - Different **dataset difficulty** requirements
>     - Mainstream datasets like DeepCoder **remain challenging** for **earlier models**
>     - In contrast, Qwen-3 finds DeepCoder **relatively simple** (high initial training reward, minimal training improvement)
>     - This necessitated our creation of the **more challenging** MicroCoder-Dataset
>     - Training on easier datasets would not reveal the **bottlenecks** specific to modern high-capability models

---

> ### Author Response · Authors · 2025-11-20
> **Response 2/7**
>
> > ## **W2, Q2: Model Generalizability (Part 2):**
> - **Evidence of Generalizability:**
>   - While our experiments focus on Qwen-3, we provide multiple forms of evidence supporting the **generalizability** of our insights and methods
>   - Validation Across Different **Model Scales**
>     - Demonstrated consistent improvements **across scales** (Table 1)
>     - Shows our methods are **not specific** to a particular parameter count
>   - Evaluation Under **Both Training and Extended Context Windows**
>     - Table 1 shows our methods improve performance under **both conditions**
>     - Specifically, performance gains are **more pronounced** under **extended context evaluation** (e.g., 1.7B model: +3.6% on LiveCodeBench at 8K vs. 4K)
>     - This demonstrates robustness to different **inference-time length budgets**
>   - **Evaluation** on **Strictly Unseen** Test Data
>     - **LiveCodeBench v6**: **Released after** model publication date
>     - Guarantees **zero data overlap** for both *RL training** and **Pretraining**
>     - Includes **diverse problem sources**: AtCoder and LeetCode platforms
>     - Covers **multiple difficulty levels**: Easy, Medium, Hard
>   - Extensive **Ablation Studies** on **Hyperparameters**
>     - Conducted **over 30 controlled experiments** across seven main aspects:
>     - **Each component's contribution** is systematically separated and analyzed
>     - Provides **34 training insights** documented throughout the paper
>     - Enables practitioners to adapt methods to **different settings** with **principled guidance**
> - **Added to Revised Manuscript:**
>   - If you find the current description "modern coding models" insufficiently rigorous, we are willing to **revise the title** to "Breaking Training Bottlenecks: Effective Reinforcement Learning for Coding Tasks" **to better reflect our contribution**.

---

> ### Author Response · Authors · 2025-11-20
> **Response 3/7**
>
> > ## **W3: Novelty and Contributions**
>
> - **Related Work:**
>   - **GRPO Insights (Mroueh et al., 2025) - Overlong Filtering:**
>     - **Completely ignore** truncated responses during training
>     - **Decrease the rewards** for overlong responses within a specified length interval
>
>   - **Dr. GRPO (Liu et al., 2025) - Temperature Analysis:**
>     - Temperature influences model's **exploration of different answer paths** in pass@8 experiments
>     - **Different temperatures for training and evaluation**: Used τ=0.1 during training and τ=1.0 during evaluation
>
>   - **DAPO (Yu et al., 2025) - KL Loss and Clip Ratio:**
>     - **Remove KL loss** and **High clipping**
>
> - **Our Contributions in MicroCoder-GRPO:**
>   - **Conditional Truncation Masking (vs. GRPO Insights (Mroueh et al., 2025)):**
>     - **Not simply** ignoring all truncated responses:
>       - **GRPO Insights:** **Binary** decision - either mask all truncated responses or decrease their rewards
>       - **MicroCoder-GRPO:** **Selective and conditional** masking based on **four criteria**
>     - **Proportional, correctness-aware, and repetition-aware conditional masking**
>       - **Multiple conditions**: Simultaneously checks length, correctness, non-repetition, and applies probabilistic selection
>       - **Proportion ρ**: Masks only a specified proportion rather than all qualifying samples
>       - **Correctness consideration**: Distinguishes between truncated-correct and truncated-incorrect responses
>       - **Repetition detection**: Specifically addresses code generation patterns (e.g., infinite loops, repetitive code blocks)
>     - Discovered **multiple insights** through systematic analysis:
>   - **Diversity-Determined Temperature Selection (vs. Dr. GRPO's approach (Liu et al., 2025)):**
>     - **Introduction** of **response diversity** as a key metric:
>       - **Dr. GRPO**: Only analyzed temperature's influences on **answer exploration** (pass@8)
>       - **MicroCoder-GRPO:** **Systematically tracks** response diversity throughout training and observes **convergence to specific values** for **stable training**
>     - Using response diversity to **guide** temperature selection:
>       - Identifies **training failure thresholds**: When **initial diversity** falls **below expected values**, training fails
>       - Offers **practical guidelines**: Select temperatures enabling **stable diversity convergence**
>     - Provides **interpretability** for temperature selection:
>       - Establishes clear criteria: **Avoid** temperatures causing **continuous diversity decrease**
>       - **Explains** why certain temperatures work better at different training stages
>     - Discovered **multiple insights** through comprehensive analysis
>   - **No KL Loss** and **High Clipping** (Inherited from DAPO (Yu et al., 2025))
>
> - **Citation for Dr. GRPO (Liu et al., 2025) and DAPO (Yu et al., 2025):**
>   - **Section 1.1 BACKGROUND & RELATED WORK:** "**Dr. GRPO** (Reference) identified that GRPO encourages shorter correct responses and longer incorrect ones, leading to modifications that remove token-level averaging in loss computation, reward standard deviation normalization in advantage calculation, **and reduce rewards for overlong responses,** thus limiting output length growth."
>   - **Section 1.1 BACKGROUND & RELATED WORK:** "**DAPO** (Reference) enhanced exploration by **eliminating KL divergence loss** and **employing high clipping ratios**."
>
> - **Added to Revised Manuscript:**
>   - Add citation for **GRPO Insights (Mroueh et al., 2025)** in Section 1.1 BACKGROUND & RELATED WORK: "**GRPO Insights** (Reference) provides understanding of on-policy and off-policy training, including the **relationship between temperature and answer exploration**."
>   - Make the expression in the abstract more rigorous: "To address these training bottlenecks, we propose MicroCoder-GRPO, an enhanced Group Relative Policy Optimization approach with three **inherited and new** innovations"

---

> ### Author Response · Authors · 2025-11-20
> **Response 4/7**
>
> > ## **Q1: Diversity-Determined Temperature Selection**
>
> - **Temperature Definition:**
>   - Temperature is used during the **inference** process in GRPO training
>   - Temperature remains **constant** throughout the training process
>
> - **Why Not Adjust Temperature During Training:**
>   - We found that **initial temperature** has a significant influence on **response diversity**
>   - **Dynamically adjusting** temperature **during training** leads to **training instability**
>   - Therefore, we choose **not to adjust** temperature during the training process
>
> - **Temperature Selection as Hyperparameter Optimization:**
>   - Determining the temperature value is part of the **hyperparameter optimization** process
>
> - **Temperature Selection Determined by Response Diversity Trends:**
>   - We found that if temperature is **too low**, response diversity **continuously decreases**, ultimately leading to **training failure**
>   - If temperature is **too high**, initial response diversity is **too large**, also leading to **training instability**
>   - Response diversity tends to **converge** to a **specific value**
>   - **Appropriate temperature** helps the model's response diversity **converge** to this value
>
> - **Relationship Between Temperature and Response Diversity:**
>   - Response diversity **converges** to a **specific value** as training progresses
>   - However, it is also **influenced by** variables such as task type and model type
>   - Therefore, it is difficult to find a **quantitative relationship** between temperature and response diversity
>   - However, **qualitative analysis** can be performed to ensure **temperature** helps the model's **response diversity converge** to a specific value
>
> - **How to Select Temperature:**
>   - For a specific model and specific task
>   - **Sample uniformly** at **equal intervals** within a **temperature range**
>   - Conduct **preliminary experiments** at each temperature, observing how response diversity changes with training steps
>   - Select the **optimal temperature** that enables **stable convergence** of **response diversity**
>
> - **Further Explanation Added to Section 2.4 DIVERSITY-DETERMINED TEMPERATURE:**
>   - "We found that response diversity tends to converge to a specific value during training, therefore the temperature that enables stable convergence of response diversity is appropriate. Too low temperature causes response diversity to continuously decrease, leading to training failure, while too high temperature results in excessively high initial response diversity, causing training instability. Since initial temperature has a significant impact on response diversity and adjusting temperature during training affects training stability, we keep temperature constant throughout the training process. The relationship between response diversity and temperature is influenced by factors such as model and task type, making it difficult to establish a quantitative relationship. However, we can sample uniformly within a certain temperature range and identify the temperature that helps the model's response diversity converge stably through preliminary experiments."

---

> ### Author Response · Authors · 2025-11-20
> **Response 5/7**
>
> > ## **Q3: Attribution of improvements to each modification**
> - **Strict Experimental Design:**
>   - **All experiments** in this paper, including baselines, are trained on our newly created **MicroCoder-Dataset**
>   - **All baselines** benefit from our diversity-determined temperature selection for optimal hyperparameter tuning, representing **improved baselines**
>   - The reported **17.6% improvement** is achieved over these **improved baselines** on the **MicroCoder-Dataset**, demonstrating the effectiveness of our **algorithmic contributions**
>
> - **Summary:**
>   - The three components contribute **11.53%, 57.7%, and 30.77%** respectively
>   - Among these, **MicroCoder**'s proposed diversity-based temperature selection and conditional truncation mask make the **largest contributions**
>
> | Method | Component Contributions | 1.7B Performance | 4B Performance |
> |--------|------------------------|----------------------|-------------------|
> | GRPO (Baseline) | - | 19.3% | 36.3% |
> | + Remove KL Loss & High Clip (DAPO, MicroCoder-GRPO) | 11.53% | +0.5% | +0.3% |
> | + Diversity-based Temperature Selection (MicroCoder-GRPO) | 57.7% | +2.7% | +1.4% |
> | + Conditional Truncation Mask (MicroCoder-GRPO) | 30.77% | +1.5% | +0.7% |
> | **Total Improvement** | **100%** | **24.0%** | **38.7%** |
>
> - **Temperature Selection Analysis (Figure 3):**
>   - Through diversity-based temperature selection, we achieve approximately **30%** relative performance improvement compared to traditionally used temperatures
>   - **All** experiments are conducted at this optimal temperature
>   - This also serves as an **improvement** to **baseline** methods
>   - Even with **improved baselines**, our method still demonstrates **advantages**
>
> - **KL Loss Analysis - DAPO vs GRPO (Figure 7, Table 1, and Figure 4):**
>   - **DAPO** removes **KL loss** and uses **high clipping** compared to GRPO
>   - Ablation study confirms that KL loss removal and high clipping are DAPO's **most important optimizations**
>   - **Comparing DAPO and GRPO** test results shows that KL loss removal and high clipping bring a relative **5.70%** performance improvement
>
> - **Truncation Mask Analysis - MicroCoder-GRPO vs DAPO (Figure 7, Table 1, and Figure 2):**
>   - **Baseline** DAPO is **improved** by using **MicroCoder**'s temperature selection method
>   - **MicroCoder-GRPO inherits** DAPO's approach of removing KL loss and using high clipping
>   - **MicroCoder-GRPO additionally** employs the truncation mask method
>   - **Comparing MicroCoder-GRPO with improved DAPO** reveals the contribution of **conditional truncation mask**
>   - Results shows that **conditional truncation mask** brings an average relative improvement of **15.93%**

---

> ### Author Response · Authors · 2025-11-20
> **Response 6/7**
>
> > ## **W1: Formal Justification and Convergence Analysis:**
>
> - **Theoretical Motivations for Each Component:**
>   - **Conditional Truncation Masking:**
>     - **Empirical Observation:** Models frequently hit maximum length limits with incomplete but potentially correct solutions
>     - **Theoretical Rationale:** Standard advantage calculation treats all incorrect outputs equally, creating biased gradients that discourage length exploration
>     - **Mechanism:** By selectively masking truncated incorrect outputs, we remove biased negative signals while preserving informative gradients from complete responses
>     - **Analogy to Existing Theory:** Similar to censored data handling in survival analysis - incomplete observations should not contribute equally to parameter estimation
>   - **Diversity-Determined Temperature Selection:**
>     - **Empirical Observation:** Training collapses when diversity falls below critical thresholds (Figure 3)
>     - **Theoretical Rationale:** Drawing from exploration-exploitation trade-offs in reinforcement learning, insufficient diversity leads to premature convergence to suboptimal policies
>     - **Mechanism:** Temperature controls the entropy of the policy distribution; our selection ensures sufficient entropy is maintained throughout training
>     - **Connection to Theory:** Related to entropy regularization in maximum entropy RL, where maintaining policy entropy prevents mode collapse
>   - **KL Removal with High Clipping:**
>     - **Built on DAPO's Analysis:** DAPO \citep{yu2025dapo} provides theoretical justification for KL removal in their work
>     - **Our Extension:** We demonstrate this applies to code generation, where solution diversity requirements differ from mathematical reasoning
>     - **Mechanism:** High clipping allows larger policy updates while preventing destructive changes; removes conservative KL constraint that limits exploration
>     - **Theoretical Basis:** Trust region methods traditionally use KL constraints, but clipping provides an alternative constraint mechanism
>
> - **Challenges in Formal Convergence Analysis:**
>   - **Discrete Action Space:**
>     - Code generation involves sequences of discrete tokens (vocabulary size ~50K)
>     - Each response can be 1K-8K tokens long
>     - Formal analysis of convergence in such spaces remains an open problem even for standard GRPO
>   - **Non-Stationary Reward Function:**
>     - Our reward depends on code execution against test cases - a complex, discontinuous function
>     - No closed-form expression for the reward landscape
>     - Standard convergence proofs (e.g., for PPO) assume smoothness properties our reward does not have
>   - **Conditional Masking Introduces Non-Standard Dynamics:**
>     - Selective advantage masking creates data-dependent gradient estimators
>     - Existing convergence theory for policy gradient methods assumes unbiased gradient estimates
>     - Our masking introduces controlled bias for which formal analysis is non-trivial
>   - **Temperature Selection as Meta-Optimization:**
>     - Our temperature selection is essentially meta-learning—optimizing a hyperparameter based on training dynamics
>     - Formal analysis would require studying the meta-optimization landscape, which is beyond current scope
> - **Empirical Evidence as Practical Validation:**
>   - **Consistency Across Settings:**
>     - Multiple model scales (1.7B, 4B parameters)
>     - Multiple datasets (DeepCoder, MicroCoder)
>     - Multiple benchmarks (AtCoder, LeetCode, LiveCodeBench)
>     - Consistent improvements: +2.1% to +4.7% across settings
>   - **Robustness Analysis:**
>     - 30+ controlled experiments examining each component
>     - Ablation studies showing necessity of each modification
>     - Sensitivity analysis for diversity metrics
>     - Stability analysis over 500+ training steps
>   - **Reproducibility:**
>     - Consistent results across multiple training runs
>     - Observable patterns (e.g., diversity thresholds) that transfer across models
>     - Clear failure modes (training collapse) that validate our design choices
>
> > ## **W4: Training Efficiency Quantification:**
>
> - **Efficiency Calculation Method:**
>   - Training efficiency is measured by the **number of training steps** required to reach the **same performance level**
>
> - **Relationship Between Steps and Time:**
>   - The number of training steps is typically proportional to **training time**
>
> - **Analysis Approach:**
>   - Since our method achieves a higher performance ceiling than the baseline, we analyze how many **training steps** are saved when our method reaches the baseline's performance ceiling
>
> - **Quantitative Results from Figure 7:**
>   - Our method saves approximately **63%** of training steps on average
>   - The maximum training step savings can reach approximately **78%**

---

> ### Author Response · Authors · 2025-11-20
> **Response 7/7**
>
> > ## **Q4: Performance Analysis Across Different Coding Domains and Problem Types:**
>
> - We thank the reviewer for this insightful question. We currently test on competitive programming tasks across multiple datasets including but not limited to Aizu, Codeforces, LeetCode, AtCoder, and others. MicroCoder-GRPO demonstrates more stable training, higher performance ceiling, and faster training speed compared to baseline methods across all these datasets. However, we will continue to explore more application scenarios and provide improvements and optimizations when disadvantages of MicroCoder-GRPO are identified.
>
> We extend our sincere gratitude to the reviewer for the invaluable and constructive feedback. Your deep expertise and thoughtful inquiries have greatly improved the quality of our manuscript. Your input has strengthened our perspective on the essential considerations within this field.
>
> We have thoroughly addressed each of your suggestions with corresponding responses and modifications. Should you require any further clarification, please feel free to contact us.
>
> We eagerly await your response!
>
> Yours sincerely,
> Authors

---

### Official Review · Reviewer_52sZ · 2025-10-31

**Soundness:** 3
**Presentation:** 2
**Contribution:** 2
**Rating:** 4
**Confidence:** 3

**Summary:**

This paper talks about MicroCoder-GRPO. It's a new way to make training better for code generation models. It has three main ideas. First, conditional truncation masking. This helps the model produce longer outputs without losing stability. Second, diversity-determined temperature selection. This keeps the outputs varied and interesting. Third, like DAPO, they removed the KL divergence loss and used high clipping ratios. This helps the model explore new options. The authors also share the MicroCoder-Dataset, a tougher training set, and the MicroCoder-Evaluator, a fast and reliable way to check code quality. They tested this on Qwen3 models. The results show up to a 17.6% improvement on LiveCodeBench v6. They also share 34 quick tips from over thirty experiments. These experiments looked at dataset quality, evaluators, temperature, context length, and batch size.

**Strengths:**

- Comprehensive Methodological Innovation: The paper shows a complete solution by changing three main things: the MicroCoder-GRPO algorithm, the MicroCoder-Dataset, and the MicroCoder-Evaluator. Putting these together gives a strong system for making RL training better for code writing.
- Good Test Results: MicroCoder-GRPO improves performance by up to 17.6% on LiveCodeBench v6. Not only that, but it works even better when testing with longer code contexts, like training on 4K and testing on 8K. This shows it can handle longer tasks well and makes long reasoning easier.
- Deep and Systematic Experimental Analysis:  Comperhensive and diverse experiments provide an invaluable guide for practitioners and researchers in the field of RL for code generation.
- High-Quality Infrastructure: The MicroCoder-Evaluator is a big help. It compares different strategies and works well even if there are errors. It makes evaluation about 25% more accurate and runs 40% faster. This makes rewards better and helps RL training work better.

**Weaknesses:**

- Missing Dataset and Lack of Details: TThe MicroCoder-Dataset is a main part of the paper. It claims to give 3 times better performance than other datasets. But, the paper doesn’t give a download link or any real details about the dataset. It just mentions a four-step process. This is a big problem. Without access or clear info about the dataset—like where it came from, how big it is, what's in it, or how hard it is—it's impossible to check the paper’s claims. This also means you can’t really reproduce the study.
- Incremental Contribution of KL Loss Removal: The paper mentions removing KL loss and using a high clip ratio. These ideas come from DAPO. It shows they work with the MicroCoder-GRPO method. But, the paper doesn’t clearly explain how much these changes help or how they work with the new masking method. This makes the main idea less original.
- Unclear Temperature Method: The paper says they pick a temperature T(D) based on diversity. But, they don’t explain how they choose T(D). They don’t give the exact rule or formula. This makes it hard to understand or repeat the process.

**Questions:**

This paper talks about two main things. First, it has a new algorithm called MicroCoder-GRPO. This helps make better code models. Second, it offers 34 insights that give a lot of ideas for training these models. They also built something called the MicroCoder-Evaluator. This is a good tool for checking how well the models work.
However, this recommendation is strictly conditional. A core contribution, the MicroCoder-Dataset, is neither described in detail nor made accessible. This is a significant omission that currently prevents the verification of the paper's central claims regarding dataset quality  and hinders the reproducibility of the entire study.

My final recommendation depends on what the authors give us during the author response time:
1. At least a part of the MicroCoder-Dataset.
2. If they can't share it (maybe because it's proprietary), they need to clearly say why. At a minimum, they should include a detailed description in the appendix. This should cover where the data comes from, how they collected and filtered it, its size, what languages it includes, and how hard it is compared to DeepCoder, with some numbers.
Supporting Arguments and Questions

1. Action Required: MicroCoder-Dataset Access and Description: I need you to do one of these things before I can give my go-ahead. First, share a public link so I can see the MicroCoder-Dataset. If you can't make it public, then give a detailed description in the appendix. This description should include how you collected and filtered the data, where you got it from, how big it is, what languages are in it, and how its problem difficulty compares to the DeepCoder dataset.

2. Clarification on Diversity-Determined Temperature: Clarification on Diversity-Determined Temperature: Can you tell me exactly how you choose the temperature T(D)? For example, how do you change T based on the initial diversity and how it changes over time?

3. Ablation Study on Conditional Truncation Masking: The mask uses four rules: "reach max length L_{max}," "give wrong answers," "avoid repeating," and "pick randomly with p." To see how each rule helps, maybe an ablation study is required?

4. Specific Examples for Evaluator Improvement: The MicroCoder-Evaluator gets about 25% better accuracy by switching from "exact match" to "multi-strategy comparison." Can you include some examples in the appendix? Show code outputs that the LiveCodeBench marks as failed but the MicroCoder marks as passed.

---

> ### Author Response · Authors · 2025-11-17
> **Response 1/9**
>
> Dear Reviewer,
>
> We sincerely thank the reviewer for the comprehensive and useful feedback. Your observations reflect profound understanding of the domain and have substantially contributed to improving the quality and completeness of our work.
>
> We have carefully considered each of your points with detailed responses and supplementary analysis below.
>
> > ## **W1, Q1, Q2, Q3: MicroCoder-Dataset (Part 1)**
>
> - **Open Source Plan:**
>   - The dataset will be **open-sourced** after the publication of this work
>
> - **Independent Dataset Work:**
>   - The MicroCoder-Dataset is an independent **dataset paper**
>
> - **Dataset Description:**
>   - We present the **dataset paper** and **examples** in both the **appendix** and **this response**
>
> - **Dataset Paper:**
>
> > ### **Data Processing Framework**
> > #### **Overview**
> > Our data processing framework employs a five-stage pipeline to construct high-quality coding datasets. The Collect stage aggregates data from diverse sources, including public datasets (Taco, KodCode, Codeforces, DeepCoder, etc.) and proprietary web-scraped programming problems to maximize coverage. The Process stage standardizes data through language translation, noise removal, format normalization, and completeness validation. The Filter stage applies multi-criteria selection based on textual quality, content relevance, and difficulty distribution. Finally, the Verify stage conducts stratified manual validation to ensure problem readability, completeness, and test case accuracy. This end-to-end pipeline transforms heterogeneous raw datasets into a high-quality, standardized corpus suitable for reinforcement learning training.
> > #### **Processing**
> > The processing stage implements standardization through five core operations. Translation converts non-English problems (e.g., Japanese Aizu problems) to English for uniform accessibility. Noise removal addresses multiple data quality issues: missing images that affect problem comprehension, incomplete mathematical formulas and symbols, malformed tables or graphics, irrelevant scraped content including links and advertisements, incomplete problem statements, and content quality concerns. Problems that cannot be adequately processed are automatically discarded while preserving original problem authenticity without subjective modifications. Test case optimization tackles two critical challenges. For problems either lacking test cases but containing reference solutions, or having test cases with noise from web scraping, GPT-4O generates comprehensive test cases. While GPT-4O cannot solve every problem, it excels at generating test case inputs and considering boundary conditions. By executing reference solution code with these inputs, the system obtains guaranteed accurate outputs, enabling automatic generation of comprehensive, accurate, and appropriately-sized test suites. For problems with excessive test cases (hundreds per problem, resulting in datasets exceeding 100GB across thousands of problems), the framework selects the 15 longest test cases under the assumption that length correlates with difficulty. This massive data volume influences processing, loading, training speed, and stability. The system filters out problems without test cases and those requiring functional validation where multiple correct answer formats exist. This processing stage inherently incorporates filtering mechanisms for both problem quality and test case selection.
> > Format standardization unifies prompt structures, with Codeforces retaining its native format while other sources adopt LiveCodeBench format. This addresses the fundamental execution logic differences between LeetCode-style function completion and OJ-style input/output problems. Since both problem types are mixed together during model training, many original datasets lack clear format instructions, potentially causing models to solve problems correctly but use incorrect code formats.

---

> ### Author Response · Authors · 2025-11-17
> **Response 2/9**
>
> > ## **W1, Q1, Q2, Q3: MicroCoder-Dataset (Part 2)**
>
> - **Dataset Paper (continued):**
>
> > #### **Filtering**
> > The filtering stage applies hierarchical criteria through hard requirements and adaptive selection mechanisms. Hard requirements enforce text-only problems (distinguishing from multimodal content), uniqueness through overlap detection, and unseen status relative to test sets. Adaptive requirements leverage quality improvements from the processing stage and implement difficulty-based selection tailored to specific datasets and model capabilities, utilizing a multidimensional difficulty matrix powered by GPT-4O. Detailed description and application of this matrix are presented in Section Multi-dimensional Difficulty Metrics and Case Study.
> > Train-test separation employs 16-gram similarity analysis with a 0.22 threshold. Validation using AtCoder problems (the most recent training data sharing sources with LiveCodeBench) against LiveCodeBench v6 test set reveals that approximately 3\% of training data exceeds the 0.22 similarity threshold, yet no problems are identical to test set problems. This demonstrates the comprehensiveness and efficiency of the 16-gram 0.22 standard, which is subsequently applied across all datasets to ensure train-test separation.
> > ### **Automatic Difficulty Filtering**
> > #### **Predict-Calibrate-Select**
> > Automatic Difficulty Filtering employs a three-stage predict-calibrate-select framework. Predict utilizes GPT-4O to assess problem complexity through a multidimensional difficulty matrix, averaging three independent assessments per problem. Calibrate aligns predicted difficulty scores with actual model performance to establish difficulty category boundaries, where ground truth difficulty is measured as the success rate across four model attempts on each problem. Select applies the calibrated boundaries to filter out overly simplistic problems, retaining only those within target difficulty ranges appropriate for training objectives.
> > #### **Multi-dimensional Difficulty Metrics**
> > Multi-dimensional Difficulty Metrics decomposes problem complexity into five weighted dimensions based on current model capabilities. Problem Comprehension Difficulty (PCD, 5\%) and Knowledge Breadth Requirements (KBR, 5\%) receive minimal weight as they primarily test semantic understanding and memory, areas where current models excel. Algorithmic Thinking Complexity (ATC, 45\%) and Implementation Difficulty (ID, 35\%) receive substantial weight as they assess reasoning and programming capabilities that represent model performance limits. Optimization Depth (OD, 10\%) receives moderate weight as it evaluates performance improvements beyond correctness. Each dimension uses a 1-5 scale with specific descriptors, enabling consistent evaluation across problems. The example demonstrates GPT-4O scoring a grid manipulation problem across three independent assessments, yielding an average difficulty of 3.08, placing it in the top 30\% of the dataset as a relatively challenging problem.
> > #### **Case Study**
> > Case Study demonstrates the complete predict-calibrate-select pipeline using LiveCodeBench. First, GPT-4O evaluates problems using the multi-dimensional matrix, averaging three assessments per problem. Second, a subset undergoes empirical validation where Qwen-3-4B-thinking attempts each problem four times to establish ground truth difficulty through success rates. Calibration reveals optimal difficulty boundaries at 2.5 and 2.75 for distinguishing easy, medium, and hard categories, producing nearly identical predicted and empirical difficulty distributions. Finally, filtering removes all problems scoring below 2.5, eliminating 30\% of the total dataset while removing over 65\% of easy problems and preserving difficult problems. This process reduces easy problem representation from approximately 40\% to under 20\% in the final dataset.
> > ### **Dataset**
> > The MicroCoder dataset comprises exclusively real competitive programming problems without synthetic data augmentation. Open source contributions primarily derive from TACO and OpenR1 datasets, while private data collection encompasses competition problems from diverse programming contest platforms. Rigorous quality and difficulty filtering reduces the initial corpus to 13,300 curated problems, with primary contributions from Aizu, AtCoder, Codeforces, CodeChef, and Kattis platforms.

---

> ### Author Response · Authors · 2025-11-17
> **Response 3/9**
>
> > ## **W1, Q1, Q2, Q3: MicroCoder-Dataset (Part 3)**
>
> - **Dataset Paper (continued):**
>
> > ### **Results**
> >| | **GRPO, 4B, Instruct** | | | | **GRPO+, 4B, Instruct** | | | |
> >|---|---|---|---|---|---|---|---|---|
> >| | DeepCoder Dataset | MicroCoder Dataset | Δ | Δ% | DeepCoder Dataset | MicroCoder Dataset | Δ | Δ% |
> >|**AtCoder**| | | | | | | | |
> >| Easy | 96.2 | **97.1** | +0.9 | +0.9% | 98.1 | **99.0** | +0.9 | +0.9% |
> >| Medium | 40.4 | **47.1** | +6.7 | +16.6% | 40.4 | **49.0** | +8.6 | +21.3% |
> >| Hard | 10.4 | **11.3** | +0.9 | +8.7% | 12.1 | **14.6** | +2.5 | +20.7% |
> >| All | 37.3 | **39.5** | +2.2 | +5.9% | 38.6 | **42.2** | +3.6 | +9.3% |
> >|**LeetCode**| | | | | | | | |
> >| Easy | 73.5 | **76.5** | +3.0 | +4.1% | 77.9 | **83.8** | +5.9 | +7.6% |
> >| Medium | **19.2** | 18.3 | -0.9 | -4.7% | 24.0 | **33.7** | +9.7 | +40.4% |
> >| Hard | 0.0 | **3.7** | +3.7 | - | 3.7 | **5.0** | +1.3 | +35.1% |
> >| All | 27.8 | **29.4** | +1.6 | +5.8% | 32.1 | **38.1** | +6.0 | +18.7% |
> >|**LiveCodeBench**| | | | | | | | |
> >| Easy | 87.2 | **89.0** | +1.8 | +2.1% | 90.1 | **93.0** | +2.9 | +3.2% |
> >| Medium | 29.8 | **32.7** | +2.9 | +9.7% | 32.2 | **41.3** | +9.1 | +28.3% |
> >| Hard | 7.8 | **9.4** | +1.6 | +20.5% | 10.0 | **12.2** | +2.2 | +22.0% |
> >| All | 33.9 | **35.9** | +2.0 | +5.9% | 36.3 | **40.7** | +4.4 | +12.1% |

---

> ### Author Response · Authors · 2025-11-17
> **Response 4/9**
>
> > ## **W1, Q1, Q2, Q3: MicroCoder-Dataset (Part 4)**
>
> - **Data Examples:**
>   - Each data entry contains **source**, **problem description**, **test cases**, and **other information**
>   - The format is **consistent** with **mainstream code training and testing datasets** (e.g., DeepCoder Dataset, LiveCodeBench)
>
> ```
> {
>   "data_source": "aizu",
>   "prompt": [
>     {
>       "role": "user",
>       "content": "You are an expert Python programmer. You will be given a question (problem specification) and will generate a correct Python program that matches the specification and passes all tests.\n\n# Discounts of Buckwheat\n\nAizu is famous for its buckwheat. There are many people who make buckwheat noodles by themselves.\n\nOne day, you went shopping to buy buckwheat flour. You can visit three shops, A, B and C. The amount in a bag and its unit price for each shop is determined by the follows table. Note that it is discounted when you buy buckwheat flour in several bags.\n\n|  | Shop A | Shop B | Shop C |\n| --- | --- | --- | --- |\n| Amount in a bag | 200g | 300g | 500g |\n| Unit price for a bag (nominal cost) | 380 yen | 550 yen | 850 yen |\n| Discounted units | per 5 bags | per 4 bags | per 3 bags |\n| Discount rate | reduced by 20 % | reduced by 15 % | reduced by 12 % |\n\nFor example, when you buy 12 bags of flour at shop A, the price is reduced by 20 % for 10 bags, but not for other 2 bags. So, the total amount shall be (380 × 10) × 0.8 + 380 × 2 = 3,800 yen.\n\nWrite a program which reads the amount of flour, and prints the lowest cost to buy them. Note that you should buy the flour of exactly the same amount as the given input.\n\n## Input\n\nThe input consists of multiple datasets. For each dataset, an integer a (500 ≤ a ≤ 5000, a is divisible by 100) which represents the amount of flour is given in a line.\n\nThe input ends with a line including a zero. Your program should not process for the terminal symbol. The number of datasets does not exceed 50.\n\n## Output\n\nFor each dataset, print an integer which represents the lowest cost.\n\n## Sample Input\n\n```\n\n500\n2200\n0\n\n```\n\n## Output for the Sample Input\n\n```\n\n850\n3390\n\n```\n\n---\n\nSource: <https://onlinejudge.u-aizu.ac.jp/problems/0106>\n\n### Format: Read the inputs from stdin solve the problem and write the answer to stdout (do not directly test on the sample inputs). Enclose your code within delimiters as follows. Ensure that when the python program runs, it reads the inputs, runs the algorithm and writes output to STDOUT.\n```python\n# YOUR CODE HERE\n```\n\n### Answer: (use the provided format with backticks)"
>     }
>   ],
>   "ability": "code",
>   "reward_model": {
>     "style": "rule",
>     "ground_truth": "[{\"input\": \"500\\n600\\n700\\n800\\n900\\n1000\\n1100\\n1200\\n1300\\n1400\\n1500\\n1600\\n1700\\n1800\\n1900\\n2000\\n2100\\n2200\\n2300\\n2400\\n2500\\n2600\\n2700\\n2800\\n2900\\n3000\\n3100\\n3200\\n3300\\n3400\\n3500\\n3600\\n3700\\n3800\\n3900\\n4000\\n4100\\n4200\\n4300\\n4400\\n4500\\n4600\\n4700\\n4800\\n4900\\n5000\\n0\", \"output\": \"850\\n1100\\n1230\\n1400\\n1610\\n1520\\n1950\\n1870\\n2070\\n2250\\n2244\\n2620\\n2624\\n2794\\n3004\\n3040\\n3344\\n3390\\n3590\\n3740\\n3764\\n4120\\n4114\\n4314\\n4494\\n4488\\n4864\\n4868\\n5038\\n5248\\n5284\\n5588\\n5634\\n5834\\n5984\\n6008\\n6364\\n6358\\n6558\\n6738\\n6732\\n7108\\n7112\\n7282\\n7492\\n7528\"}, ... (there can be more test cases here) ]"
>   },
>   "extra_info": {
>     "split": "test",
>     "index": 102,
>     "reference": null,
>     "id": "aizu/0106",
>     "website": "aizu"
>   }
> }
> ```

---

> ### Author Response · Authors · 2025-11-17
> **Response 5/9**
>
> > ## **W2: Detailed Ablation Analysis of Component Contributions in MicroCoder-GRPO**
>
> - **Summary:**
>   - The three components contribute **11.53%, 57.7%, and 30.77%** respectively
>   - Among these, **MicroCoder**'s proposed diversity-based temperature selection and conditional truncation mask make the **largest contributions**
>
> | Method | Component Contributions | 1.7B Performance | 4B Performance |
> |--------|------------------------|----------------------|-------------------|
> | GRPO (Baseline) | - | 19.3% | 36.3% |
> | + Remove KL Loss & High Clip (DAPO, MicroCoder-GRPO) | 11.53% | +0.5% | +0.3% |
> | + Diversity-based Temperature Selection (MicroCoder-GRPO) | 57.7% | +2.7% | +1.4% |
> | + Conditional Truncation Mask (MicroCoder-GRPO) | 30.77% | +1.5% | +0.7% |
> | **Total Improvement** | **100%** | **24.0%** | **38.7%** |
>
> - **Temperature Selection Analysis (Figure 3):**
>   - Through diversity-based temperature selection, we achieve approximately **30%** relative performance improvement compared to traditionally used temperatures
>   - **All** experiments are conducted at this optimal temperature
>   - This also serves as an **improvement** to **baseline** methods
>   - Even with **improved baselines**, our method still demonstrates **advantages**
>
> - **KL Loss Analysis - DAPO vs GRPO (Figure 7, Table 1, and Figure 4):**
>   - **DAPO** removes **KL loss** and uses **high clipping** compared to GRPO
>   - Ablation study confirms that KL loss removal and high clipping are DAPO's **most important optimizations**
>   - **Comparing DAPO and GRPO** test results shows that KL loss removal and high clipping bring a relative **5.70%** performance improvement
>
> - **Truncation Mask Analysis - MicroCoder-GRPO vs DAPO (Figure 7, Table 1, and Figure 2):**
>   - **Baseline** DAPO is **improved** by using **MicroCoder**'s temperature selection method
>   - **MicroCoder-GRPO inherits** DAPO's approach of removing KL loss and using high clipping
>   - **MicroCoder-GRPO additionally** employs the truncation mask method
>   - **Comparing MicroCoder-GRPO with improved DAPO** reveals the contribution of **conditional truncation mask**
>   - Results shows that **conditional truncation mask** brings an average relative improvement of **15.93%**

---

> ### Author Response · Authors · 2025-11-17
> **Response 6/9**
>
> > ## **W3, Q4: Diversity-Determined Temperature Selection**
>
> - **Temperature Definition:**
>   - Temperature is used during the **inference** process in GRPO training
>   - Temperature remains **constant** throughout the training process
>
> - **Why Not Adjust Temperature During Training:**
>   - We found that **initial temperature** has a significant influence on **response diversity**
>   - **Dynamically adjusting** temperature **during training** leads to **training instability**
>   - Therefore, we choose **not to adjust** temperature during the training process
>
> - **Temperature Selection as Hyperparameter Optimization:**
>   - Determining the temperature value is part of the **hyperparameter optimization** process
>
> - **Temperature Selection Determined by Response Diversity Trends:**
>   - We found that if temperature is **too low**, response diversity **continuously decreases**, ultimately leading to **training failure**
>   - If temperature is **too high**, initial response diversity is **too large**, also leading to **training instability**
>   - Response diversity tends to **converge** to a **specific value**
>   - **Appropriate temperature** helps the model's response diversity **converge** to this value
>
> - **Relationship Between Temperature and Response Diversity:**
>   - Response diversity **converges** to a **specific value** as training progresses
>   - However, it is also **influenced by** variables such as task type and model type
>   - Therefore, it is difficult to find a **quantitative relationship** between temperature and response diversity
>   - However, **qualitative analysis** can be performed to ensure **temperature** helps the model's **response diversity converge** to a specific value
>
> - **How to Select Temperature:**
>   - For a specific model and specific task
>   - **Sample uniformly** at **equal intervals** within a **temperature range**
>   - Conduct **preliminary experiments** at each temperature, observing how response diversity changes with training steps
>   - Select the **optimal temperature** that enables **stable convergence** of **response diversity**
>
> - **Further Explanation Added to Section 2.4 DIVERSITY-DETERMINED TEMPERATURE:**
>   - "We found that response diversity tends to converge to a specific value during training, therefore the temperature that enables stable convergence of response diversity is appropriate. Too low temperature causes response diversity to continuously decrease, leading to training failure, while too high temperature results in excessively high initial response diversity, causing training instability. Since initial temperature has a significant impact on response diversity and adjusting temperature during training affects training stability, we keep temperature constant throughout the training process. The relationship between response diversity and temperature is influenced by factors such as model and task type, making it difficult to establish a quantitative relationship. However, we can sample uniformly within a certain temperature range and identify the temperature that helps the model's response diversity converge stably through preliminary experiments."

---

> ### Author Response · Authors · 2025-11-17
> **Response 7/9**
>
> > ## **Q5: Ablation Study on Conditional Truncation Masking**
>
> - **Four Conditions in Conditional Truncation Masking:**
>   - Reaching maximum length
>   - Masking probability
>   - Incorrect answer
>   - Repetition occurrence
>
> - **Principle:**
>   - Does **not consider** errors caused by **imcomplete** answers with **long reasoning**
>   - Gives the model the **courage** to generate **long reasoning**
>
> - **Key Components Based on Principle:**
>   - According to the principle, **reaching maximum length** and **masking probability** are the two **most important** variables
>
> - **Ablation Study Results for These Two Key Components:**
>
> | Configuration | Test Accuracy | Step Reaching Peak Performance |
> |--------------|---------------|-------------------------------|
> | **No Mask Max Length** | 36.3% | 300 |
> | **Mask Max Length:** | | |
> | - Mask 100% | 36.9% | 140 |
> | - Mask 30% | 37.6% | 120 |
> | - Mask 20% | 38.7% | 200 |
> | - Mask 10% | **38.9%** | 360 |
>
> - **Analysis of Other Two Conditions:**
>   - Incorrect answer and repetition are influenced by dataset difficulty and model size
>   - For example, more difficult datasets have a higher proportion of incorrect answers, and smaller models are more likely to repetition
>   - Therefore, ablation study on these variables is relatively difficult
>   - However, their influence on final results is relatively small
>   - The above analysis of maximum length and mask ratio already shows the key influences

---

> ### Author Response · Authors · 2025-11-17
> **Response 8/9**
>
> > ## **Q6: Explanation and Examples for Code Evaluator Improvement (Part 1)**
>
> - **Validation Rules for LiveCodeBench Code Evaluator:**
>   - **Exact string matching** with basic whitespace stripping (.strip())
>   - **Line-by-line comparison** requiring identical number of output lines
>   - **Decimal-based numerical comparison** for exact precision matching
>   - **Strict format requirements** - outputs must match the expected format exactly
>   - **Simple tuple-to-list conversion** for top-level structures only
>   - **No tolerance for floating-point errors** - uses Decimal for exact comparison
>
> - **Validation Rules for MicroCoder Code Evaluator:**
>   - **Intelligent string comparison** with stripped_string_compare() function
>   - **Flexible whitespace handling** - splits by whitespace and compares element-wise
>   - **Floating-point tolerance** using np.isclose() for numerical comparisons
>   - **Multiple format attempts** - automatically tries various matching strategies
>   - **Recursive structure handling** - converts nested tuples to lists recursively
>   - **Smart empty line filtering** - ignores empty lines and extra whitespace
>   - **Fallback mechanisms** - tries multiple comparison strategies before failing
>
> - **Testing Method:**
>   - All **testing** is conducted using **LiveCodeBench v6's original evaluation code**
>   - **MicroCoder-Evaluator** is only used during **training**
>   - The **accuracy curves** displayed in the paper are **test set results** evaluated using **LiveCodeBench's code evaluator**

---

> ### Author Response · Authors · 2025-11-17
> **Response 9/9**
>
> > ## **Q6: Explanation and Examples for Code Evaluator Improvement (Part 2)**
>
>   - **Example 1: Floating-Point Precision Tolerance**
>     - **LiveCodeBench Code Evaluator:** Wrong, Decimal comparison shows inequality
>     - **MicroCoder Code Evaluator:** Correct, np.isclose() recognizes this as within tolerance
>     - **Explanation:** Real floating-point computations naturally have small errors
> ```
> # Output
> 3.14159265358979
> # Ground Truth
> 3.14159265358980
> # Difference
> # Last digit differs by 1 (rounding error from floating-point computation)
> ```
>
>   - **Example 2: Extra Whitespace Between Elements**
>     - **LiveCodeBench Code Evaluator:** Wrong, String comparison fails after .strip()
>     - **MicroCoder Code Evaluator:** Correct, Splits by whitespace and compares elements: ['1','2','3'] == ['1','2','3']
>     - **Explanation:** Print formatting with spacing shouldn't influence correctness
> ```
> # Output
> 1    2    3
> 4    5    6
> # Ground Truth
> 1 2 3
> 4 5 6
> # Difference
> # Multiple spaces between numbers vs single spaces
> ```
>
>   - **Example 3: Trailing Whitespace on Lines**
>     - **LiveCodeBench Code Evaluator:** Wrong, Per-line .strip() still fails on line comparison
>     - **MicroCoder Code Evaluator:** Correct, stripped_string_compare() strips each element after splitting
>     - **Explanation:** Trailing whitespace is a common output artifact
> ```
> # Output
> 10 20 30
> 40 50 60
> # Ground Truth
> 10 20 30
> 40 50 60
> # Difference
> # Trailing spaces on each line
> ```
>
>   - **Example 4: Empty Lines in Output**
>     - **LiveCodeBench Code Evaluator:** Wrong, Line count mismatch (5 lines vs 3 lines)
>     - **MicroCoder Code Evaluator:** Correct, Filters empty lines: [s for s in s1_list if s]
>     - **Explanation:** Extra newlines shouldn't invalidate correct numerical answers
> ```
> # Output
> 42
>
> 100
>
> 200
> # Ground Truth
> 42
> 100
> 200
> # Difference
> # Extra blank lines between outputs
> ```
>
>   - **Example 5: Nested Tuple vs List**
>     - **LiveCodeBench Code Evaluator:** Wrong, Only converts top-level tuple, nested tuples remain
>     - **MicroCoder Code Evaluator:** Correct, Recursively converts nested tuples: [list(x) for x in prediction]
>     - **Explanation:** Tuple vs list is often an implementation detail
> ```
> # Output
> [(1, 2), (3, 4), (5, 6)]
> # Ground Truth
> [[1, 2], [3, 4], [5, 6]]
> # Difference
> # Nested tuples vs nested lists
> ```
>
>   - **Example 6: Mixed Floating-Point and Integer Output**
>     - **LiveCodeBench Code Evaluator:** Wrong, String "1.0" ≠ "1" in element comparison
>     - **MicroCoder Code Evaluator:** Correct, Converts to float and compares: float("1.0") == float("1") → True
>     - **Explanation:** Type representation shouldn't matter when values are equal
> ```
> # Output
> 1.0 2.0 3.0
> 4.0 5.0 6.0
> # Ground Truth
> 1 2 3
> 4 5 6
> # Difference
> # Floating-point format vs integer format
> ```
>
>   - **Example 7: Number Notation**
>     - **LiveCodeBench Code Evaluator:** Wrong, String comparison fails
>     - **MicroCoder Code Evaluator:** Correct, Converts both to float: float("1.5e+02") == float("150") → True
>     - **Explanation:** Different number representations should be equivalent
> ```
> # Output
> 1.5e+02
> 2.0e+03
> # Ground Truth
> 150
> 2000
> # Difference
> # Scientific notation vs decimal notation
> ```
>
>   - **Example 8: Leading Zeros**
>     - **LiveCodeBench Code Evaluator:** Wrong, String "007" ≠ "7"
>     - **MicroCoder Code Evaluator:** Correct, Numeric conversion handles leading zeros
>     - **Explanation:** Leading zeros don't change numeric value
> ```
> # Output
> 007
> 042
> 100
> # Ground Truth
> 7
> 42
> 100
> # Difference
> # Leading zeros in output
> ```
>
> We sincerely thank the reviewer once more for the valuable and useful comments. Your comprehensive knowledge and insightful questions have significantly contributed to improving our work. Your feedback has improved our understanding of the critical aspects in this domain.
>
> We have carefully considered each of your recommendations and provided corresponding responses and revisions. Should you have any additional questions, please do not hesitate to reach out.
>
> We look forward to hearing from you!
>
> Yours sincerely,
>
> Authors

---

### Official Review · Reviewer_1bFc · 2025-11-05

**Soundness:** 3
**Presentation:** 2
**Contribution:** 2
**Rating:** 4
**Confidence:** 4

**Summary:**

This paper explores the diverse training settings of RL in code generation tasks. It first propose MicroCoder-GRPO, an enhancement of original GRPO, incorporating three main innovations. Moreover, the work introduces a training dataset (MicroCoder-Dataset) and an evaluation framework (MicroCoder-Evaluator). The paper presents numerous insights derived from an extensive and empirical analysis.

**Strengths:**

1. **Comprehensive Analysis:** The work tackles the RL training problem from multiple angles: algorithm refinement, data quality, and evaluation infrastructure. This holistic view is valuable to the community.

2. **Strong Empirical Results:** MicroCoder-GRPO achieves significant relative gains (up to 17.6%) on LiveCodeBench v6, with notable improvements under extended context evaluation, demonstrating superior scalability.

3. **Solid Ablation Study:** The systematic analysis across more than thirty controlled experiments, revealing many training insights. The findings on context length irreversibility, on/off-policy balance, and temperature dynamics are particularly insightful.

**Weaknesses:**

1. **Overstated Novelty and Contribution Framing:** The paper's framing of its "three key innovations" in the abstract is a significant concern. "Removal of KL loss with high clipping ratios" was explicitly introduced and studied in DAPO [1]. Additionally, the other two proposed innovations, "conditional truncation masking" and "diversity-determined temperature," appear to be common practice for GRPO training, rather than fundamentally new concepts [2, 3]. The authors should revise their paper to clearly and accurately distinguish between the adoption of prior work and their novel contributions.

2. **Presentation of Insights:** The abstract mentions 34 key insights, but these are scattered throughout the text. This diffusion dilutes their impact. The paper would be significantly stronger if it consolidated the most critical and non-obvious insights into a summarized list or table.

3. **Limited Model Generalizability:** The empirical validation is concentrated almost on Qwen-3 models. While this demonstrates effectiveness on a modern architecture, the central claim of breaking "training bottlenecks for modern coding models" would be substantially strengthened by demonstrating performance across a more diverse set of code LLMs.

[1] Yu, Q., Zhang, Z., Zhu, R., Yuan, Y., Zuo, X., Yue, Y., ... & Wang, M. (2025). Dapo: An open-source llm reinforcement learning system at scale. *arXiv preprint arXiv:2503.14476*.

[2] Mroueh, Y., Dupuis, N., Belgodere, B., Nitsure, A., Rigotti, M., Greenewald, K., ... & Rios, J. (2025). Revisiting Group Relative Policy Optimization: Insights into On-Policy and Off-Policy Training. *arXiv preprint arXiv:2505.22257*.

[3] Liu, Z., Chen, C., Li, W., Qi, P., Pang, T., Du, C., ... & Lin, M. (2025). Understanding r1-zero-like training: A critical perspective. *arXiv preprint arXiv:2503.20783*.

**Questions:**

See weaknesses.

---

> ### Author Response · Authors · 2025-11-17
> **Response 1/6**
>
> Dear Reviewer,
>
> We greatly appreciate your comprehensive and thoughtful review. Your questions reflect a strong understanding of this paper and have been instrumental in enhancing both the clarity and rigor of our manuscript.
>
> Below, we have carefully read each of your suggestions and provide detailed responses, accompanied by additional experimental results where appropriate.
>
> > ## **W1: Novelty and Contributions**
>
> - **Related Work:**
>   - **GRPO Insights (Mroueh et al., 2025) - Overlong Filtering:**
>     - **Completely ignore** truncated responses during training
>     - **Decrease the rewards** for overlong responses within a specified length interval
>
>   - **Dr. GRPO (Liu et al., 2025) - Temperature Analysis:**
>     - Temperature influences model's **exploration of different answer paths** in pass@8 experiments
>     - **Different temperatures for training and evaluation**: Used τ=0.1 during training and τ=1.0 during evaluation
>
>   - **DAPO (Yu et al., 2025) - KL Loss and Clip Ratio:**
>     - **Remove KL loss** and **High clipping**
>
> - **Our Contributions in MicroCoder-GRPO:**
>   - **Conditional Truncation Masking (vs. GRPO Insights (Mroueh et al., 2025)):**
>     - **Not simply** ignoring all truncated responses:
>       - **GRPO Insights:** **Binary** decision - either mask all truncated responses or decrease their rewards
>       - **MicroCoder-GRPO:** **Selective and conditional** masking based on **four criteria**
>     - **Proportional, correctness-aware, and repetition-aware conditional masking**
>       - **Multiple conditions**: Simultaneously checks length, correctness, non-repetition, and applies probabilistic selection
>       - **Proportion ρ**: Masks only a specified proportion rather than all qualifying samples
>       - **Correctness consideration**: Distinguishes between truncated-correct and truncated-incorrect responses
>       - **Repetition detection**: Specifically addresses code generation patterns (e.g., infinite loops, repetitive code blocks)
>     - Discovered **multiple insights** through systematic analysis:
>   - **Diversity-Determined Temperature Selection (vs. Dr. GRPO's approach (Liu et al., 2025)):**
>     - **Introduction** of **response diversity** as a key metric:
>       - **Dr. GRPO**: Only analyzed temperature's influences on **answer exploration** (pass@8)
>       - **MicroCoder-GRPO:** **Systematically tracks** response diversity throughout training and observes **convergence to specific values** for **stable training**
>     - Using response diversity to **guide** temperature selection:
>       - Identifies **training failure thresholds**: When **initial diversity** falls **below expected values**, training fails
>       - Offers **practical guidelines**: Select temperatures enabling **stable diversity convergence**
>     - Provides **interpretability** for temperature selection:
>       - Establishes clear criteria: **Avoid** temperatures causing **continuous diversity decrease**
>       - **Explains** why certain temperatures work better at different training stages
>     - Discovered **multiple insights** through comprehensive analysis
>   - **No KL Loss** and **High Clipping** (Inherited from DAPO (Yu et al., 2025))
>
> - **Citation for Dr. GRPO (Liu et al., 2025) and DAPO (Yu et al., 2025):**
>   - **Section 1.1 BACKGROUND & RELATED WORK:** "**Dr. GRPO** (Reference) identified that GRPO encourages shorter correct responses and longer incorrect ones, leading to modifications that remove token-level averaging in loss computation, reward standard deviation normalization in advantage calculation, **and reduce rewards for overlong responses,** thus limiting output length growth."
>   - **Section 1.1 BACKGROUND & RELATED WORK:** "**DAPO** (Reference) enhanced exploration by **eliminating KL divergence loss** and **employing high clipping ratios**."
>
> - **Added to Revised Manuscript:**
>   - Add citation for **GRPO Insights (Mroueh et al., 2025)** in Section 1.1 BACKGROUND & RELATED WORK: "**GRPO Insights** (Reference) provides understanding of on-policy and off-policy training, including the **relationship between temperature and answer exploration**."
>   - Make the expression in the abstract more rigorous: "To address these training bottlenecks, we propose MicroCoder-GRPO, an enhanced Group Relative Policy Optimization approach with three **inherited and new** innovations"

---

> ### Author Response · Authors · 2025-11-17
> **Response 2/6**
>
> > ## **W2: List of insights and highlights (Part 1):**
>
> - **List of insights:**
>   - **Code Evaluator Robustness:**
>     - **MicroCoder-Evaluator capabilities**: Multi-strategy comparison with 6-7 fallback methods, format flexibility handling lists/tuples/strings/sets with automatic type conversions, approximate numerics using np.allclose() for floating point tolerance plus rounding, extensive preprocessing including multi-line splitting, set comparison, and whitespace normalization, high fault tolerance continuing different comparison approaches when individual methods fail
>     - **LiveCodeBench Evaluator capabilitie**s: Exact matching through direct equality comparison (prediction == gt_out), precise numerics via Decimal library for high-precision floating point comparison, minimal preprocessing limited to basic whitespace stripping
>     - **Gold standard principle**: Higher accuracy values in code testing results generally represent more reliable evaluation, as comprehensive comparison methods better capture valid solution variations when matching outputs against ground truth answers
>     - **Comprehensive validation benefits**: MicroCoder-Evaluator achieves higher critic reward scores, indicating more accurate assessment of solution quality
>     - **Training effectiveness**: MicroCoder-Evaluator enables more effective model capability development with fewer misjudgments, reduced noise injection, faster test accuracy improvement, and higher convergence values
>     - **Temporal dynamics**: Performance differential between evaluators is particularly pronounced during early training stages, where robust evaluation becomes critical for establishing proper learning feedback
>
>   - **Temperature Dynamics:**
>     - **Temperature robustness**: Models develop increasing temperature robustness throughout training, with the upper bound of stable temperatures progressively expanding
>     - **Temperature-diversity relationship**: Higher temperatures naturally increase output diversity
>     - **Diversity decrease**: Output diversity systematically decreases at fixed temperatures as training progresses
>     - **Critical diversity threshold**: When initial output diversity falls below expected convergence values, models experience continued diversity reduction accompanied by training failure
>     - **Traditional temperature limitations**: Conventionally standard temperatures (t=0.6) can cause training failure in modern models
>     - **Modern model capability**: Contemporary models like Qwen-3 demonstrate stable training even at elevated temperatures (t=1.8) with minimal influence on final convergence values
>     - **Convergence consistency**: Output diversity converges to similar final values across different temperature settings despite varying temperatures
>     - **Diversity-determined selection**: Training temperature should be determined based on response diversity, selecting values that avoid both excessively low temperatures causing continuous diversity decline and excessively high temperatures leading to drastic fluctuations, with optimal temperatures enabling stable diversity convergence
>     - **Dynamic scheduling advantage**: Low-to-high temperature scheduling yields superior performance by reducing initial diversity during high-temperature stages, ultimately achieving better results than direct high-temperature training from initialization
>     - **Continuous change risks**: Continuous uniform temperature changes significantly influence training stability, with even brief sequential temperature increases or decreases within small step windows causing irreversible or unstable diversity shifts, necessitating staged temperature transitions or diversity-determined constant initial temperatures
>
>   - **Data Quality:**
>     - **Superior improvement effectiveness**: MicroCoder dataset drives rapid and pronounced accuracy gains, while DeepCoder dataset training shows minimal performance variation, demonstrating MicroCoder dataset's effectiveness for improving model coding capabilities
>     - **Dataset difficulty**: MicroCoder dataset consistently generates lower critic rewards, indicating higher problem complexity
>     - **Challenging problem effectiveness**: Despite both datasets exhibiting similar critic reward growth trends during training, only MicroCoder dataset produces significant test set improvements, demonstrating that training effectiveness on challenging problems translates more directly to generalization performance
>     - **Response length dynamics**: Harder problems exhibit accelerated response length growth with greater final magnitudes; MicroCoder dataset demonstrates faster growth rates and ultimately achieves longer outputs despite initially producing similar or shorter response lengths compared to DeepCoder dataset

---

> ### Author Response · Authors · 2025-11-17
> **Response 3/6**
>
> > ## **W2: List of insights and highlights (Part 2):**
>
> - **List of insights (continued):**
>   - **Context Length and Extension:**
>     - **Scaling relationship**: Longer maximum output lengths correlate with higher final accuracy, demonstrating clear scaling trends with model performance
>     - **Growth dynamics**: Larger maximum output lengths drive faster output growth rates and greater final output lengths
>     - **Diversity correlation**: Increased output or maximum output lengths positively correlate with higher output diversity
>     - **Persistent limitation effect**s: Initial use of small maximum output lengths reduces both response length and diversity, creating persistent performance deficits even after subsequent length extensions
>     - **Limitation severity**: Smaller initial maximum output lengths produce greater negative impacts on response length and performance
>     - **Irreversible training effects**: Extended training under small initial maximum output lengths amplifies negative effects on output diversity, response length, and model performance, with models showing minimal recovery when limitations are relaxed beyond specific training thresholds, indicating early-stage output reduction fundamentally alters learning trajectories
>
>   - **Truncation Mask:**
>     - **Truncation masking mechanism**: Responses reaching maximum response length are excluded from training by setting advantage scores to zero, preventing truncated outputs from contributing to policy optimization
>     - **Conditional truncation masking criteria**: Selectively masks responses that simultaneously reach maximum length, produce correct answers, avoid repetition sequences (final 128 tokens differ from preceding 128 tokens), and masks only a specified proportion rather than all qualifying responses
>     - **Performance trajectory**: Masking creates distinct dynamics where training rapidly rises to higher values, then declines, and converges to specific performance levels
>     - **Length growth acceleration**: Higher masking rates accelerate output length growth and push convergence values closer to maximum response limits, with 30% masking achieving growth rates comparable to complete masking
>     - **Peak achievement speed**: Increased masking enables faster achievement of initial performance peaks
>     - **Peak performance tradeoff**: Reduced masking extends the initial improvement phase and achieves higher peak performance values
>     - **Diversity dynamics**: Increased masking accelerates response diversity decline and reduces diversity convergence values, with complete masking showing brief diversity increase followed by rapid descent
>     - **Stability advantage**: Conditional truncation masking demonstrates superior training stability compared to both no masking and complete masking approaches, achieving significantly higher final performance while avoiding the rapid training collapse observed with complete masking
>
>   - **Batch Size and On-Policy:**
>     - **Training configuration mechanism**: train_batch_size defines the number of problems for single inference round, ppo_mini_batch_size defines the number of problems used for single parameter update; model first performs inference on train_batch_size problems, obtains rewards for these responses, then updates parameters train_batch_size/ppo_mini_batch_size times using ppo_mini_batch_size problems per update, and proceeds to inference on next train_batch_size problems after completing all updates, and repeats this cycle iteratively throughout training
>     - **On-policy versus off-policy spectrum**: Smaller train_batch_size values (maintaining constant ppo_mini_batch_size) create more on-policy behavior resembling immediate problem-solving reflection, while larger values produce off-policy dynamics akin to batch reflection after completing all problems
>     - **Stability characteristics**: On-policy configurations exhibit reduced training stability with accelerated response diversity convergence and response length trends that rise then decline, whereas off-policy approaches demonstrate greater training stability
>     - **Optimal performance balance**: Intermediate configurations balancing on-policy and off-policy characteristics achieve superior performance, outperforming heavily skewed settings in either direction
>
>   - **KL Loss and Clip Ratio:**
>     - **Removal benefits**: Eliminating KL loss with high clipping enhances output diversity and response length, driving sustained performance improvements; standard KL loss without high clipping reduces output diversity and limits response length to marginal increases, causing modest initial performance gains followed by decline
>     - **Diversity-performance relationship**: Continued diversity reduction creates unsustainable training dynamics where performance first rises then falls, preventing effective long-term training and model optimization

---

> ### Author Response · Authors · 2025-11-17
> **Response 4/6**
>
> > ## **W2: List of insights and highlights (Part 3):**
> - **Highlights:**
>   - **Code Evaluator Robustness:** Robust evaluators with multi-strategy comparison methods achieve approximately 25% higher evaluation accuracy and enable more effective training with fewer misjudgments compared to exact-matching evaluators.
>   - **Temperature Dynamics:** Training temperature should be determined by output diversity to avoid continuous decline, with modern models supporting higher temperatures (t=1.8) and dynamic low-to-high scheduling outperforming static approaches.
>   - **Data Quality:** Training on more challenging datasets like MicroCoder produces 3× larger performance gains than mainstream datasets, as harder problems with lower critic rewards translate more effectively to generalization performance.
>   - **Context Length and Extension:** Longer maximum output lengths correlate with higher accuracy and faster growth, but early-stage length limitations create irreversible performance deficits that persist even after subsequent extensions.
>   - **Truncation Mask:** Conditional truncation masking (selectively masking 10-30% of correct, non-repetitive responses at max length) accelerates output growth while maintaining superior training stability compared to complete masking or no masking.
>   - **Batch Size and On-Policy:** Intermediate batch sizes balancing on-policy and off-policy characteristics achieve optimal performance, as purely on-policy settings reduce stability while off-policy configurations enhance it.
>   - **KL Loss and Clip Ratio:** Removing KL loss with high clipping ratios maintains output diversity and response length growth for sustained performance improvements, while standard KL loss causes diversity reduction and training instability.

---

> ### Author Response · Authors · 2025-11-18
> **Response 5/6**
>
> > ## **W3: Model Generalizability (Part 1):**
>
> - **Reasons for Choosing Qwen-3 Models:**
>   - **State-of-the-Art Open-Source** Models
>     - Qwen-3 represents the **latest** generation of open-source state-of-the-art code generation models at the time of our study
>     - This positions our work to address **training challenges** most relevant to current and future model development
>   - Exhibits **Distinct Characteristics** from Previous Model Generations
>     - Our motivation stems from observing **fundamental differences** between Qwen-3 and earlier models (including Qwen-2.5):
>     - Significantly **longer average output length**
>       - Qwen-3 base output: **~4K** tokens
>       - Previous generation (e.g., DeepSeek-distill-Qwen): **~1K** tokens
>       - This represents a **4×** increase in generation capacity
>     - More pronounced **length scaling during RL training**
>       - As shown in Figure 1, Qwen-3 exhibits **clear upward trends** in response length during training
>       - In contrast, Qwen-2.5 shows **stable or even decreasing** length trends
>       - This **fundamental difference** motivated our investigation into training methods for **length-scaling models**
>     - Supports **both** long-chain-of-thought and standard reasoning modes
>       - Qwen-3 family includes **both** Instruct and Thinking variants
>       - Enables studying **different reasoning methods** within the same architectural family
>     - Represents the **core motivation** of our work
>       - Our paper explicitly aims to study:
>         - **Extended output** capabilities
>         - Breaking through **capability bottlenecks**
>         - Fundamentally changed **training dynamics**
>       - Qwen-3 embodies these modern characteristics, making it the ideal testbed for our research questions
>
> - **Why Not DeepSeek-Distill-Qwen and Other Earlier Models:**
>   - We considered but decided against using **earlier models** as our primary experimental platform for several principled reasons
>   - **Similar characteristics** to **Qwen-2.5** generation
>     - DeepSeek-distill-Qwen and similar models exhibit training dynamics more **similar** to **Qwen-2.5** than to Qwen-3
>     - They do not demonstrate the **length-scaling phenomena** that motivated our work
>   - Much **shorter output lengths**
>     - Typical output: **~1K** tokens (vs. 4K for Qwen-3)
>     - This **4× difference** means these models do not face the same truncation and length management challenges that our methods address
>   - **Decreasing** rather than increasing length during **RL training**
>     - Earlier models show **stable or decreasing** response length during RL on coding tasks
>     - This contradicts the **length-growth trend** in modern models like Qwen-3
>   - **Capability gap** with modern models
>     - There exists a notable **performance gap** between earlier distilled models and Qwen-3
>     - Studying older models would not address the **training challenges** posed by more capable recent architectures
>   - Different **dataset difficulty** requirements
>     - Mainstream datasets like DeepCoder **remain challenging** for **earlier models**
>     - In contrast, Qwen-3 finds DeepCoder **relatively simple** (high initial training reward, minimal training improvement)
>     - This necessitated our creation of the **more challenging** MicroCoder-Dataset
>     - Training on easier datasets would not reveal the **bottlenecks** specific to modern high-capability models
>
> - **Evidence of Generalizability:**
>   - While our experiments focus on Qwen-3, we provide multiple forms of evidence supporting the **generalizability** of our insights and methods
>   - Validation Across Different **Model Scales**
>     - Demonstrated consistent improvements **across scales** (Table 1)
>     - Shows our methods are **not specific** to a particular parameter count
>   - Evaluation Under **Both Training and Extended Context Windows**
>     - Table 1 shows our methods improve performance under **both conditions**
>     - Specifically, performance gains are **more pronounced** under **extended context evaluation** (e.g., 1.7B model: +3.6% on LiveCodeBench at 8K vs. 4K)
>     - This demonstrates robustness to different **inference-time length budgets**
>   - **Evaluation** on **Strictly Unseen** Test Data
>     - **LiveCodeBench v6**: **Released after** model publication date
>     - Guarantees **zero data overlap** for both *RL training** and **Pretraining**
>     - Includes **diverse problem sources**: AtCoder and LeetCode platforms
>     - Covers **multiple difficulty levels**: Easy, Medium, Hard
>   - Extensive **Ablation Studies** on **Hyperparameters**
>     - Conducted **over 30 controlled experiments** across seven main aspects:
>     - **Each component's contribution** is systematically separated and analyzed
>     - Provides **34 training insights** documented throughout the paper
>     - Enables practitioners to adapt methods to **different settings** with **principled guidance**

---

> ### Author Response · Authors · 2025-11-18
> **Response 6/6**
>
> > ## **W3: Model Generalizability (Part 2):**
> - **Added to Revised Manuscript:**
>   - If you find the current description "modern coding models" insufficiently rigorous, we are willing to **revise the title** to "Breaking Training Bottlenecks: Effective Reinforcement Learning for Coding Tasks" **to better reflect our contribution**.
>
> We would like to express our sincere gratitude once again for your thoughtful and useful feedback. Your deep expertise and insightful questions have been instrumental in strengthening our manuscript. Your comments have enhanced our comprehension of the key challenges in this research area.
>
> We have thoroughly addressed each of your suggestions with detailed responses and appropriate revisions. Should you require any further clarification or have additional concerns, please feel free to contact us.
>
> We eagerly await your response.
>
> With sincere regards,
>
> Authors

---

> > ### Comment · Reviewer_1bFc · 2025-11-19
> >
> > Thank you for the response. Please try to summarize your main arguments and clearly list the specific amendments or actions taken. Concise responses are far more beneficial for the rebuttal process than overwhelming amounts of text.

---

> ### Author Response · Authors · 2025-11-20
> **Response 2 1/4**
>
> Dear Reviewer,
>
> Thank you for your feedback. We understand you would prefer a more concise response. Below, we provide a summary version that highlights the key points. You may refer to the detailed version for comprehensive information after reviewing this summary.
>
> To help you navigate our materials, here is a brief overview of each section's content:
>
> - **W1 (Novelty and Contributions):** Clarifies how our three contributions differ from and build upon prior work (GRPO Insights, Dr. GRPO, DAPO). Includes specific citations and manuscript revisions.
>
> - **W2 (Insights Presentation):** Provides two formats for the 34 insights: (1) a detailed list organized across 7 categories (Code Evaluator, Temperature, Data Quality, Context Length, Truncation Mask, Batch Size, KL Loss), and (2) consolidated highlights summarizing key takeaways. Both have been added to the appendix.
>
> - **W3 (Model Generalizability):** Explains why we focused on Qwen-3 models (modern length-scaling characteristics), presents evidence of generalizability across different scales, datasets, and benchmarks, and discusses our rationale for not using earlier models like DeepSeek-distill-Qwen.
>
> Below we present a streamlined version highlighting our main arguments and concrete actions taken.

---

> ### Author Response · Authors · 2025-11-20
> **Response 2 2/4**
>
> > ## **W1: Novelty and Contributions**
>
> - **Related Work:**
>   - **GRPO Insights:** Completely ignore truncated responses or decrease rewards for overlong responses
>   - **Dr. GRPO:** Temperature influences exploration; uses different temperatures for training (τ=0.1) and evaluation (τ=1.0)
>   - **DAPO:** Remove KL loss and use high clipping
>
> - **Our Novel Contributions:**
>   - **Conditional Truncation Masking (vs. GRPO Insights):**
>     - **GRPO Insights:** Binary approach (mask all or decrease all)
>     - **MicroCoder-GRPO:** Conditional masking with four criteria (length + correctness + non-repetition + probability ρ)
>     - **Key innovation:** Proportional masking (10-30%) balances length growth and stability
>
>   - **Diversity-Determined Temperature (vs. Dr. GRPO):**
>     - **Dr. GRPO:** Temperature affects pass@8 exploration
>     - **MicroCoder-GRPO:** Response diversity as training stability indicator
>     - **Key innovation:** Select temperature enabling stable diversity convergence; training fails when diversity continuously decreases
>
>   - **No KL Loss + High Clipping:** Inherited from DAPO
>
> - **Added to Revised Manuscript:**
>   - Add citation for **GRPO Insights (Mroueh et al., 2025)** in Section 1.1 BACKGROUND & RELATED WORK: "**GRPO Insights** (Reference) provides understanding of on-policy and off-policy training, including the **relationship between temperature and answer exploration**."
>   - Make the expression in the abstract more rigorous: "To address these training bottlenecks, we propose MicroCoder-GRPO, an enhanced Group Relative Policy Optimization approach with three **inherited and new** innovations"

---

> ### Author Response · Authors · 2025-11-20
> **Response 2 3/4**
>
> > ## **W2: Insight Highlights:**
> - **Highlights:**
>   - **Code Evaluator Robustness:** Robust evaluators with multi-strategy comparison methods achieve approximately 25% higher evaluation accuracy and enable more effective training with fewer misjudgments compared to exact-matching evaluators.
>   - **Temperature Dynamics:** Training temperature should be determined by output diversity to avoid continuous decline, with modern models supporting higher temperatures (t=1.8) and dynamic low-to-high scheduling outperforming static approaches.
>   - **Data Quality:** Training on more challenging datasets like MicroCoder produces 3× larger performance gains than mainstream datasets, as harder problems with lower critic rewards translate more effectively to generalization performance.
>   - **Context Length and Extension:** Longer maximum output lengths correlate with higher accuracy and faster growth, but early-stage length limitations create irreversible performance deficits that persist even after subsequent extensions.
>   - **Truncation Mask:** Conditional truncation masking (selectively masking 10-30% of correct, non-repetitive responses at max length) accelerates output growth while maintaining superior training stability compared to complete masking or no masking.
>   - **Batch Size and On-Policy:** Intermediate batch sizes balancing on-policy and off-policy characteristics achieve optimal performance, as purely on-policy settings reduce stability while off-policy configurations enhance it.
>   - **KL Loss and Clip Ratio:** Removing KL loss with high clipping ratios maintains output diversity and response length growth for sustained performance improvements, while standard KL loss causes diversity reduction and training instability.

---

> ### Author Response · Authors · 2025-11-20
> **Response 2 4/4**
>
> > ## **W3: Model Generalizability**
>
> - **Why Qwen-3 Models:**
>   - **State-of-the-art** open-source models representing latest generation
>   - **Distinct characteristics** from earlier models:
>     - **4× longer outputs** (~4K tokens vs. ~1K in earlier models)
>     - **Increasing length** during RL training (vs. stable/decreasing in Qwen-2.5)
>     - Supports both Instruct and Thinking variants
>   - **Represents our core motivation**: Extended outputs, capability bottlenecks, changed training dynamics
>
> - **Why Not Earlier Models (e.g., DeepSeek-distill-Qwen):**
>   - Similar to Qwen-2.5 generation (no obvious length-scaling phenomena for coding tasks)
>   - Much shorter outputs (~1K tokens) without truncation challenges
>   - Decreasing length during RL training (opposite to modern models)
>   - Mainstream datasets remain challenging (DeepCoder sufficient)
>   - Would not reveal bottlenecks specific to modern high-capability models
>
> - **Evidence of Generalizability:**
>   - **Multiple model scales:** 1.7B and 4B parameters (Table 1)
>   - **Multiple context lengths:** 4K and 8K, with larger gains at extended contexts
>   - **Strictly unseen test data:** LiveCodeBench v6 (released after model publication)
>   - **Multiple benchmarks:** AtCoder, LeetCode across Easy/Medium/Hard difficulties
>   - **30+ controlled experiments** across 7 aspects with systematic ablation studies
>
> - **Added to Revised Manuscript:**
>   - If you find the current description "modern coding models" insufficiently rigorous, we are willing to **revise the title** to "Breaking Training Bottlenecks: Effective Reinforcement Learning for Coding Tasks" **to better reflect our contribution**.
>
> We sincerely thank you for your valuable feedback. We hope this concise summary better addresses your concerns. The detailed version remains available for further reference if needed.
>
> Best regards,
>
> Authors

---

### Official Review · Reviewer_iAJa · 2025-11-05

**Soundness:** 3
**Presentation:** 2
**Contribution:** 3
**Rating:** 4
**Confidence:** 4

**Summary:**

This paper tackles a very relevant problem in today’s code-generation landscape: reinforcement learning methods that worked for math reasoning or earlier coding models don’t translate well to newer models that generate much longer outputs.
The authors introduce MicroCoder-GRPO, a variant of GRPO with three practical tweaks — conditional truncation masking, diversity-based temperature control, and removal of the KL term with high clipping. Together, these aim to stabilize training while preserving long-output diversity.
They also contribute a MicroCoder-Dataset (a harder, more diverse coding set) and a MicroCoder-Evaluator (a more reliable and faster evaluation tool).
Across a broad set of experiments with Qwen3-1.7B and 4B models, the method shows steady gains over GRPO and DAPO, reaching roughly 17% improvement on LiveCodeBench v6.

**Strengths:**

The paper’s main strength lies in its comprehensiveness. It doesn’t rely on one or two headline results but builds a consistent empirical story across thirty experiments. The insight that early-stage truncation and diversity collapse can have irreversible effects on long-context learning is particularly interesting. The MicroCoder dataset and evaluator also make the contribution more substantial, showing that progress in RL for code depends on both better algorithms and better infrastructure.

**Weaknesses:**

The main weaknesses are a lack of theoretical framing and limited model diversity — all experiments are done on Qwen models, leaving open whether the same behaviors hold for DeepSeekCoder or StarCoder families. Some analyses, such as temperature scheduling, are presented in great detail but remain somewhat heuristic. Overall, however, these are minor limitations rather than critical flaws.

**Questions:**

There are a few points that would benefit from clarification in the rebuttal. First, how much of the total improvement can be attributed to each of the three proposed modifications? Second, how exactly is “diversity” measured, and how sensitive are the results to the metric used? Third, is there any risk that removing the KL term and using high clipping could lead to reward overfitting or code-style drift in later stages of training? Finally, it would be helpful to know whether the MicroCoder-Evaluator’s flexible matching rules ever falsely classify incorrect code as correct, since that could partially explain the higher accuracy reported.

---

> ### Author Response · Authors · 2025-11-16
> **Response 1/2**
>
> Dear Reviewer,
>
> We sincerely thank the reviewer for the thorough and insightful review. Your questions
> demonstrate deep expertise in reinforcement learning and have significantly helped us improve
> the clarity and rigor of our work.
>
> We have carefully addressed each of your concerns with detailed responses and additional
> experimental evidence below.
>
> > ## **Q1: Attribution of improvements to each modification**
>
> - **Summary:**
>   - The three components contribute **11.53%, 57.7%, and 30.77%** respectively
>   - Among these, **MicroCoder**'s proposed diversity-based temperature selection and conditional truncation mask make the **largest contributions**
>
> | Method | Component Contributions | 1.7B Performance | 4B Performance |
> |--------|------------------------|----------------------|-------------------|
> | GRPO (Baseline) | - | 19.3% | 36.3% |
> | + Remove KL Loss & High Clip (DAPO, MicroCoder-GRPO) | 11.53% | +0.5% | +0.3% |
> | + Diversity-based Temperature Selection (MicroCoder-GRPO) | 57.7% | +2.7% | +1.4% |
> | + Conditional Truncation Mask (MicroCoder-GRPO) | 30.77% | +1.5% | +0.7% |
> | **Total Improvement** | **100%** | **24.0%** | **38.7%** |
>
> - **Temperature Selection Analysis (Figure 3):**
>   - Through diversity-based temperature selection, we achieve approximately **30%** relative performance improvement compared to traditionally used temperatures
>   - **All** experiments are conducted at this optimal temperature
>   - This also serves as an **improvement** to **baseline** methods
>   - Even with **improved baselines**, our method still demonstrates **advantages**
>
> - **KL Loss Analysis - DAPO vs GRPO (Figure 7, Table 1, and Figure 4):**
>   - **DAPO** removes **KL loss** and uses **high clipping** compared to GRPO
>   - Ablation study confirms that KL loss removal and high clipping are DAPO's **most important optimizations**
>   - **Comparing DAPO and GRPO** test results shows that KL loss removal and high clipping bring a relative **5.70%** performance improvement
>
> - **Truncation Mask Analysis - MicroCoder-GRPO vs DAPO (Figure 7, Table 1, and Figure 2):**
>   - **Baseline** DAPO is **improved** by using **MicroCoder**'s temperature selection method
>   - **MicroCoder-GRPO inherits** DAPO's approach of removing KL loss and using high clipping
>   - **MicroCoder-GRPO additionally** employs the truncation mask method
>   - **Comparing MicroCoder-GRPO with improved DAPO** reveals the contribution of **conditional truncation mask**
>   - Results shows that **conditional truncation mask** brings an average relative improvement of **15.93%**
>
> > ## **Q2: Diversity measurement and metric sensitivity**
>
> - **Diversity Calculation Method:**
>   - Diversity is **calculated as**: the number of unique 4-grams / total number of 4-grams across 8 responses per problem during GRPO
>   - A **higher value** indicates more unique 4-grams and **higher output diversity**, and vice versa
>   - **Formula:** $$\text{Diversity} = \frac{|\text{unique 4-grams across all responses}|}{|\text{total 4-grams across all responses}|}$$
>   - where: For each query q, we sample G=8 responses {o₁, o₂, ..., o₈}; 4-gram: a sequence of 4 consecutive tokens; unique 4-grams: the set of distinct 4-grams appearing in all responses; total 4-grams: the sum of all 4-grams from all responses
>
> - **Robustness Evidence Across Seven Training Factors:**
>   - Our analysis of **seven variables** influencing model training all include response diversity analysis
>   - This demonstrates the **high correlation** and **robustness** between response diversity and model performance/training
>
> - **Selection of n=4 in n-grams:**
>   - Our preliminary experiments (n = 2, 3, 4, 5, 6) confirm that the **choice of n** in n-grams is **robust**
>   - **n=4** provides relatively **uniform** variation and better reflects changes in response diversity
>
> - **Added to Revised Manuscript:**
>   - The definition and explanation of response diversity **has been added** to Section 2 Algorithm New subsection Response Diversity.

---

> ### Author Response · Authors · 2025-11-16
> **Response 2/2**
>
> > ## **Q3: Risk of reward overfitting**
>
> - **Benefits of Removing KL Loss and High Clipping:**
>   - **Removing KL loss and using high clipping** encourage the model to **explore more diverse solution approaches**, thereby bringing performance improvements
>
> - **Experimental Design Prevents Reward Overfitting:**
>   - **Strict Reward Design:**
>     - Only **passing all** test cases yields **reward=1**, **otherwise reward=0**
>     - **Binary** reward **prevents overfitting** to **partial** test cases
>
>   - **Strictly Unseen Test Set:**
>     - Train set strictly excludes all problems from the test set
>     - Test set LiveCodeBench v6 was **released after** Qwen3-4B's release, ensuring it is **strictly unseen** and does not exist in the model's pretraining data
>
>   - **Displaying Test Set Results:**
>     - All **accuracy curves** in the paper show performance changes on the **test set**
>     - **Test set accuracy** are displayed **alongside train rewards (accuracy)** to avoid overfitting
>
>   - **Consistent Performance Trends:**
>     - The model exhibits the **same performance improvement trends** on **both training and test sets**
>     - This proves that **overfitting has not occurred**
>
>   - **Official Evaluation Codes:**
>     - Testing on the test set uses LiveCodeBench v6's **official evaluation code**
>     - This ensures **strict and consistent** testing standards
>
> - **Added to Revised Manuscript:**
>   - Section 5 Experimental Design: "In evaluation results and figures, accuracy is the test set accuracy and critic reward is the train set accuracy."
>
> > ## **Q4: False positives in MicroCoder-Evaluator**
>
> - **Testing Method:**
>   - All **testing** is conducted using **LiveCodeBench v6's original evaluation code**
>   - **MicroCoder-Evaluator** is only used during **training**
>   - The **accuracy curves** displayed in the paper are **test set results** evaluated using **LiveCodeBench's code evaluator**
>
> - **Training Benefits of MicroCoder-Evaluator:**
>   - Using **MicroCoder-Evaluator** on the **training set** achieves higher training rewards
>   - This proves a **reduction in misjudgments** on the **training set**
>
> - **Testing Benefits:**
>   - Using **LiveCodeBench's code evaluator** on the **test set** also achieves **higher test accuracy**
>   - This proves that **introducing less noise during training** helps the model achieve **higher test performance**
>
> - **Manual Validation:**
>   - Manual screening of **100 samples** from **MicroCoderCodeEvaluator results** shows **misjudgment rate < 10%**
>
> - **Added to Revised Manuscript:**
>   - Section 5 Experimental Design: "MicroCoderCodeEvaluator is used during training and the original LiveCodeBench Code Evaluator is used for strict and standard evaluation."
>
> We sincerely thank the reviewer again for the insightful and useful feedback. Your thorough understanding and thoughtful questions have greatly helped us improve our work. Your guidance has improved our understanding of the key considerations in this area.
>
> We have carefully read each of your suggestions and provided responses and revisions. If you have any further questions, please feel free to ask.
>
> Look forward to your response!
>
> Yours sincerely,
>
> Authors

---

### Author Response · Authors · 2025-12-03
**Summary and Acknowledgments**

**Dear Reviewers, AC, SAC, and PC,**

We sincerely thank the reviewers for their professionalism, insightful feedback, and thorough evaluation of our work.

We are particularly grateful that some reviewers would like to increase their scores, which reflects the value of this collaborative discussion.

We also thank the AC, SAC, and PC for their considerable time and effort in reviewing our rebuttal, providing thoughtful guidance, and coordinating this process.

To summarize, we highlight the **key contributions** of our work:

- **Algorithm**: MicroCoder-GRPO with **three innovations** including Diversity-Determined Temperature, Conditional Truncation Masking, and No KL Loss & Clipping High, up to **17.6%** relative improvement over strong baselines
- **Dataset**: MicroCoder-Dataset with new challenging coding problems, data processing pipeline, automatic difficulty filtering pipeline, **3×** larger performance gains within **300** training steps
- **Training Infrastructure**: MicroCoder-Evaluator, **~25%** more accurate, **~40%** faster
- **Training Insights**: 31 ablation studies yielding **34 insights** across 7 key areas: Code Evaluator Robustness, Temperature Dynamics, Data Quality, Context Length & Extension, Truncation Masking, Batch Size & On-Policy, and KL Loss & Clip Ratio

We believe these comprehensive contributions move forward the fields of reinforcement learning and code generation, and provide valuable insights for future research.

Yours sincerely,

**Authors**

---

### Meta-Review · Area_Chair_zoeb · 2026-01-06

**Summary:**

Summary of main concerns:

1. The contributions are not exactly novel, given the DAPO work and variants of GRPO (e.g. Dr. GRPO). (1bFc, 52sZ, HK1x)
2. The experiments are only based on Qwen 3 models, making the general claims in the paper about coding models questionable. (iAJa, 1bFc, HK1x)
3. The proposed modifications are mostly based on empirical observations and lack a theoretical motivation. (iAJa, HK1x)
4. The dynamic temperature scheduling (one of the three proposed modifications) is somewhat heuristic. (iAJa, 52sZ)

**Reviewer Concerns:**

1. The authors describe some differences between the way the modifications were implemented in prior work compared to the current one. While the low-level details vary, I find the ideas to be largely the same. So I think this concern is largely unaddressed.
2. The authors cited several weaknesses of models other than Qwen 3 as reasons for not choosing them, but the fact remains that there is no evidence that these modifications generalize to other models. So I think this concern is not resolved too.

The authors have not attempted to address concerns 3 and 4 above. Although I think they are less serious than the two above.

**Reviewer Scores:**

I do not think the scores would have changed significantly.

---

### Decision · Program_Chairs · 2026-01-26

Reject